# Non-canonical Wnt signaling regulates neural stem cell quiescence during homeostasis and after demyelination

Manideep Chavali[1,2,4], Michael Klingener[1], Alexandros G. Kokkosis[1], Yury Garkun[3], Sylwia Felong[1], Arianna Maffei[3] & Adan Aguirre[1]

Adult neural stem cells (NSCs) reside in a specialized microenvironment, the subventricular zone (SVZ), which provides them with unique signaling cues to control their basic properties and prevent their exhaustion. While the signaling mechanisms that regulate NSC lineage progression are well characterized, the molecular mechanisms that trigger the activation of quiescent NSCs during homeostasis and tissue repair are still unclear. Here, we uncovered that the NSC quiescent state is maintained by Rho-GTPase Cdc42, a downstream target of non-canonical Wnt signaling. Mechanistically, activation of Cdc42 induces expression of molecules involved in stem cell identity and anchorage to the niche. Strikingly, during a demyelination injury, downregulation of non-canonical Wnt-dependent Cdc42 activity is necessary to promote activation and lineage progression of quiescent NSCs, thereby initiating the process of tissue repair.

[1] Department of Pharmacological Sciences, Stony Brook University, Stony Brook, NY 11794, USA. [2] Materials Science and Engineering, Stony Brook University, Stony Brook, NY 11794, USA. [3] Department of Neurobiology and Behavior, Stony Brook University, Stony Brook, NY 11794, USA. [4] Present address: Eli & Edythe Broad Center of Regeneration Medicine & Stem Cell Research, University of California–San Francisco, San Francisco, CA 94143, USA. Michael Klingener and Alexandros G. Kokkosis contributed equally to this work. Correspondence and requests for materials should be addressed to A.A. (email: adan.aguirre@stonybrook.edu)

In the adult tissues, stem cells reside in specialized microenvironments, called "niches". Although stem cells have the highest potential to generate distinct progeny, they are themselves slowly cycling (quiescent) in adulthood, and through this behavior they regulate the maintenance of tissue homeostasis and regeneration throughout life. In the adult brain stem cell niche, the subventricular zone (SVZ), neural stem cells (NSCs, type B cells) generate intermediate transit-amplifying neural progenitors (NPCs, type C cells) that are the primary source of both glial and neuronal lineages[1,2]. After embryonic generation, a subpopulation of NSCs in the SVZ remain quiescent (qNSCs) unless activated[3,4], which can be triggered by ablation of neural cell lineages and pathological conditions in adulthood[5,6]. In the adult SVZ niche, qNSCs are found in the ventricular wall while activated NSCs (aNSCs) are found in the periventricular region[7,8], suggesting that unique cues in those microenvironments tightly regulate the positional identity of quiescent and activated NSCs. In this context, recent studies have implicated vascular cell adhesion molecule (VCAM-1) and N-Cadherin in maintaining the positioning of qNSCs in the apical niche, and disruption of these molecules induced their activation[9,10]. Similarly, alpha6 and beta1 integrins and the inhibitor of differentiation proteins (Ids) have been implicated in promoting the vascular apposition of NSCs in the basal SVZ niche[11,12]. Altogether, these recent findings revealed that the quiescent and activated states of NSCs are precisely regulated in the niche. However, the signaling cues maintaining qNSC positioning within the niche during physiology, and the molecular mechanisms that trigger the activation of qNSCs during pathological conditions to promote tissue regeneration, are still largely unclear. In this study, we show that the non-canonical Wnt pathway plays a crucial role in maintaining the quiescent status of NSCs during both normal and pathological conditions. Non-canonical Wnt signaling, through the activation of Rho-GTPase Cdc42, maintains NSC adhesion to the apical niche and regulates Notch signaling activity. Intriguingly, during a demyelination injury, downregulation of the non-canonical Wnt/Cdc42 axis and activation of canonical Wnt/β-catenin signaling in SVZ NSCs is required to achieve tissue homeostasis and repair. Our novel findings establish that a transient shift from non-canonical to canonical Wnt signaling is critical for the activation and lineage progression of qNSCs, and to achieve post-injury repair at a functional level.

## Results

### Proteomics analysis of the SVZ during demyelination.
Injury in adult tissues often recapitulates certain developmental processes. Although many of the signaling mechanisms are still active, the cellular response to injury in the adult organs is quite different from normal tissue growth. Nevertheless, development and wound repair share many common features such as temporally regulated lineage progression, cell migration, angiogenesis, and reorganization of the microenvironment[13–16]. Therefore, well-established animal models of brain injury represent a suitable approach to elucidate signaling mechanisms involved in maintenance of stemness, and more importantly to understand the cellular and molecular events involved in activation of quiescent NSCs[13–16]. In this study, we employed a mouse model of demyelination/remyelination to gain insight into the niche dynamics and molecular mechanisms that govern activation of qNSCs, and to understand how these processes impact tissue regeneration and function. A proteomics-based screen of SVZ tissue collected at the peak of chronic demyelination yielded 790 proteins that were upregulated by 1.2-fold and 166 proteins that were downregulated by 0.8-fold as compared to control SVZ[17].

Using these hits, we performed Gene Ontology (GO) analysis and detected alterations in GO categories for cell adhesion, cellular response to stimulus, and metabolic activity, all of which are indicative of NSC activation and lineage progression (Fig. 1a, b). Therefore, we next performed pathway enrichment analysis of both upregulated and downregulated proteins in the SVZ during demyelination to determine the mechanism of NSC activation, which revealed changes in the regulation of cell cycle dynamics and cell metabolism. Furthermore, along with changes in previously identified mechanisms active during demyelination such as EGFR, FGFR, and PDGF signaling[17–19], we also identified alterations in Rho-GTPase-mediated cell adhesion, Wnt signaling, and the Sonic Hedgehog pathways (Fig. 1c, Supplementary Data 1). An increase in the activity of these molecular events has been reported to be involved in activation of normally quiescent stem cells[20]. Indeed, it has also recently been reported that molecules involved in Rho-GTPase and cell adhesion-mediated signaling mechanisms are highly enriched in GFAP[+]CD133[+]EGFR[−] quiescent NSCs[5], and that they also express Wnt receptors and downstream Wnt effectors[5,21]. To build off these findings, we performed STRING protein–protein interaction analysis[22] on the proteins identified as a part of the Wnt and Rho-GTPase pathways, and observed enriched interactions between these two elements (Fig. 1d). Specifically, Cdc42 and β-catenin are among the most highly interacting proteins in this network (Fig. 1d; highlighted by red text). Furthermore, our proteomics screen identified pathway-specific proteins such as ARHGAP1, ARHGAP44, and Pak1–2 involved in the post-translational regulation of Cdc42, Rac, and Rho, and proteins such as β-catenin which are involved in mediation of both Wnt signaling and cell adhesion (Fig. 1d, e, Supplementary Data 1). Indeed, our proteomics screen also identified proteins encoded by genes that were enriched in quiescent NSCs (Supplementary Fig. 1a)[5]. Stem cell adhesion and positioning within the niche are important determinants of quiescence and activation. Given the importance of Rho-GTPase signaling in the maintenance of cell adhesion and the role of non-canonical Wnt signaling in regulating Rho-GTPases and cell polarity[23–26], we focused on the intersection of these two pathways as a candidate mechanism involved in regulating NSC identity and positioning within the SVZ niche, with emphasis on how they are altered during white matter injury and repair.

### Wnt signaling elements in the SVZ and choroid plexus.
Stem cell niches, including the SVZ microenvironment, are known to provide a Wnt-rich environment[14,27–31]. However, the functional role of canonical and non-canonical Wnt signaling in controlling the cell cycle status of NSCs is not fully understood. To this end, we first characterized all canonical and non-canonical Wnt ligand expression levels in the SVZ and choroid plexus as a first step in elucidating their potential role in regulating NSC biology. As the qNSC cilia are bathed in choroid plexus-secreted cerebrospinal fluid, we reasoned that it is likely to be a rich source of environmental signals that regulate NSC properties. Therefore, we performed qPCR analysis on both SVZ and choroid plexus-derived cDNA. Consistent with previous reports[28,32], we detected higher expression levels of canonical Wnt ligands in the SVZ as compared to choroid plexus (Fig. 2a, Supplementary Data 2). Wnt8b and Wnt9a were the only exceptions, as those were enriched in the choroid plexus. The canonical Wnt ligands Wnt3a, Wnt8a, and Wnt10a transcripts were not detectable in either of the tissues. Wnt7a and Wnt7b, both known activators of canonical Wnt/β-catenin signaling[33,34], were expressed at high levels compared to other ligands in the SVZ (Fig. 2a). Of all the non-canonical Wnt ligands, Wnt4, Wnt5a, and Wnt11 were

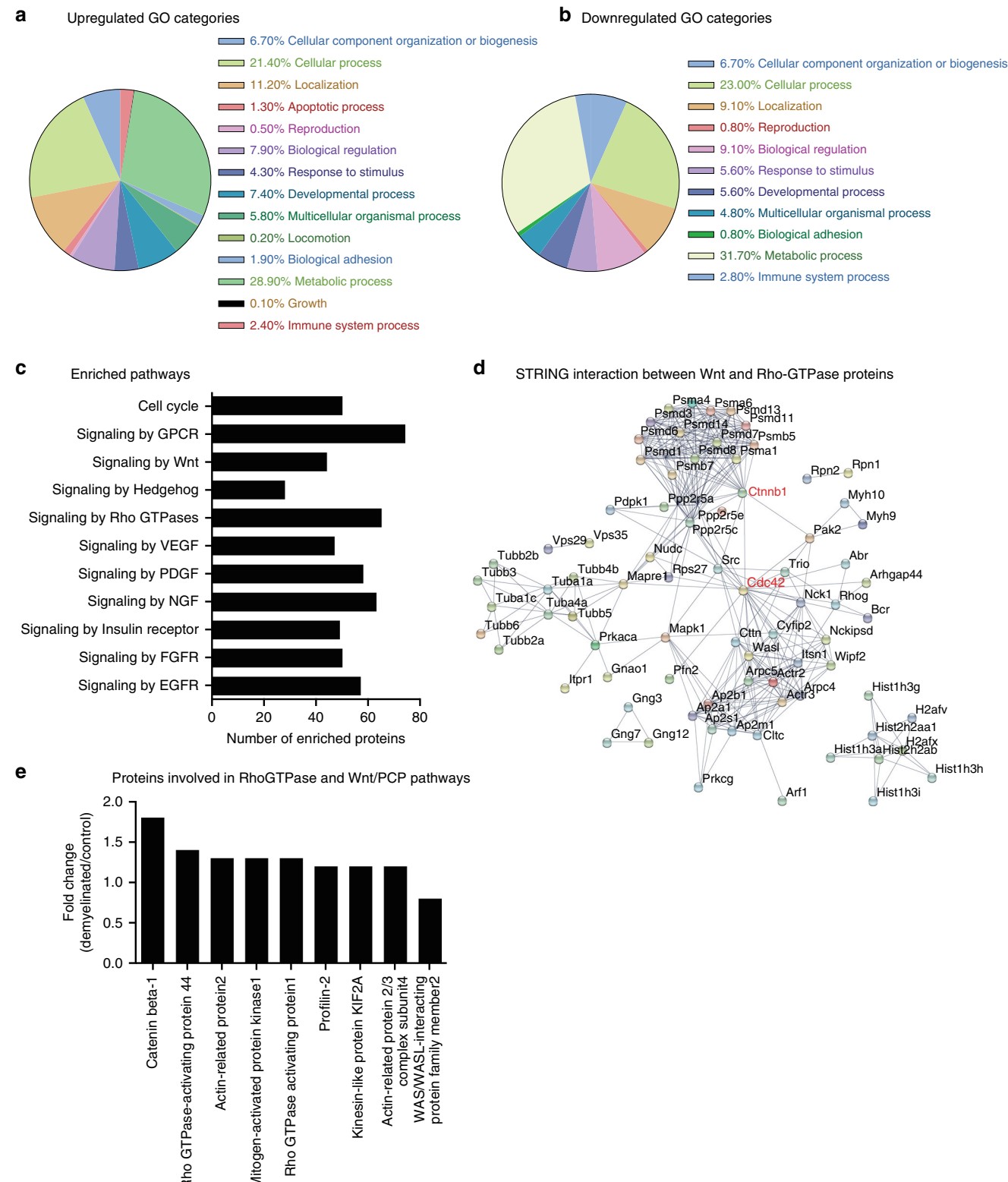

**Fig. 1** Proteomics analysis of the SVZ at the peak of demyelination. **a**, **b** Gene ontology categories of **a** upregulated and **b** downregulated proteins in the SVZ at the peak of demyelination. In total, we identified 790 proteins that were upregulated by at least 1.2-fold and 166 proteins that were downregulated by 0.8-fold. **c** Reactome pathway enrichment analysis using both upregulated and downregulated proteins identified from the proteomics screen. **d** STRING protein–protein interaction network analysis indicates enriched interactions between Wnt and Rho-GTPase signaling elements identified in the proteomics screen. Ctnnb1 (β-Catenin) and Cdc42 (both highlighted in red color) are among the most highly interacting proteins. **e** Proteins involved in the Rho-GTPase and Wnt/PCP pathways that are altered in the SVZ following a demyelination injury

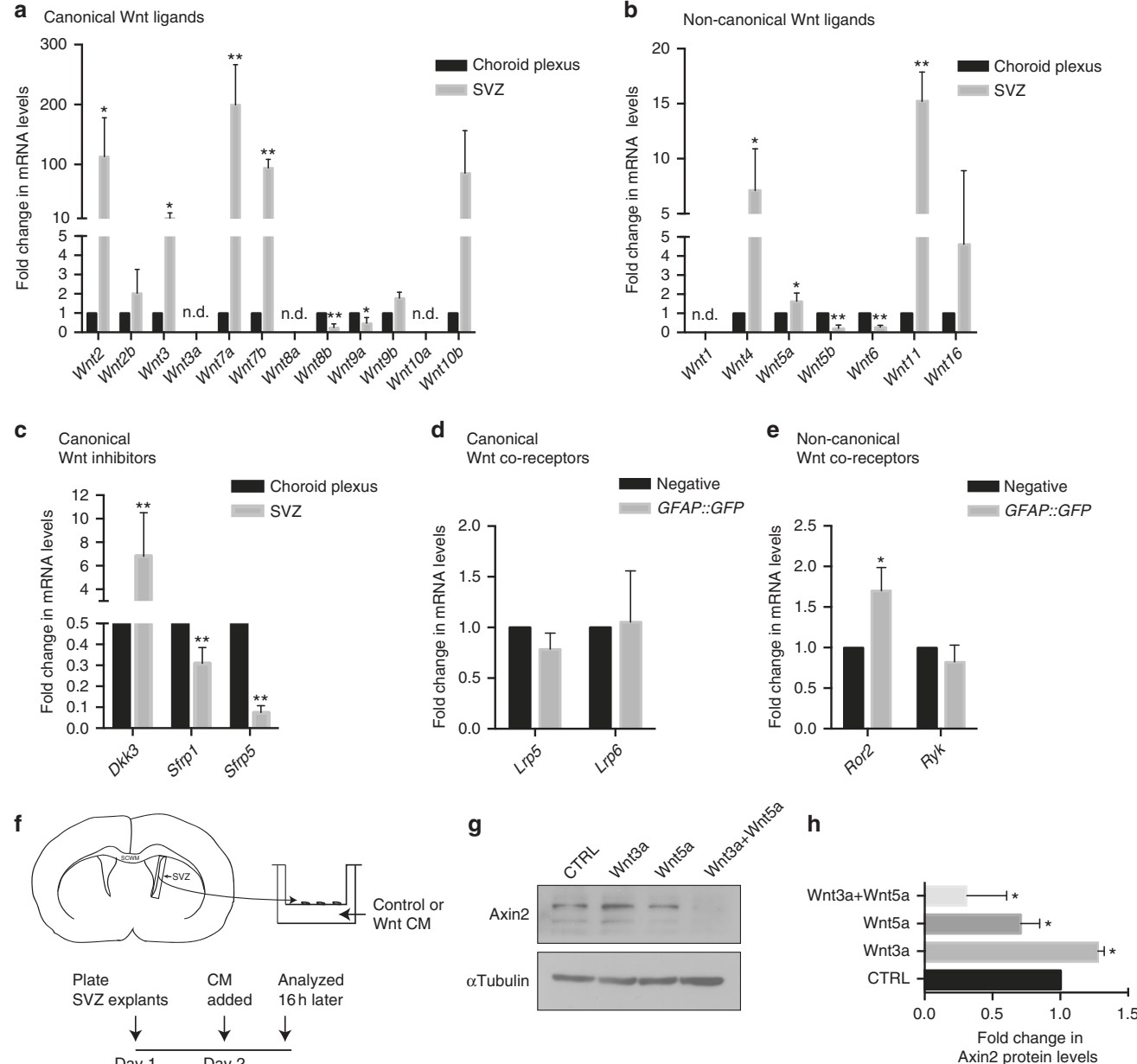

**Fig. 2** Differential expression of canonical and non-canonical Wnt pathway elements in the SVZ and choroid plexus. **a**, **b** qPCR analysis of **a** canonical and **b** non-canonical Wnt ligands generated from reverse-transcribed RNA isolated from the SVZ and choroid plexus tissues of adult mice. Transcript levels of genes in the SVZ are represented as fold change over the levels of the same genes in the choroid plexus (see also Supplementary Data 2). **c** Transcript levels of canonical Wnt/β-catenin signaling inhibitors in the SVZ and choroid plexus. **d**, **e** SVZ neural stem cells express Wnt co-receptors involved in both **d** canonical and **e** non-canonical pathways. Transcript levels of *Lrp5, Lrp6, Ror2,* and *Ryk* in sorted *GFAP::GFP*[+] cells are represented as fold change over the transcript levels of the same genes in the GFP-negative cells (see also Supplementary Figure 1b). **f** Schematic showing the SVZ explant culture system and the experimental paradigm for Wnt ligand stimulation and analysis. **g** Axin2 protein expression in SVZ explants stimulated with Control, Wnt3a, Wnt5a, or Wnt3a+Wnt5a conditioned media. **h** Densitometry quantification of Axin2 protein levels (shown in **g**; Axin2 protein levels are normalized to α-tubulin levels and represented as a fold change over the control condition). *$P < 0.05$, **$p < 0.01$; error bars represent mean ± s.e.m.; $n = 4$ for **a**–**e**; $n = 3$ for **h**; $n =$ independent experiments; n.d. not detected

enriched in the SVZ, while *Wnt5b* and *Wnt6* were expressed at higher levels in the choroid plexus (Fig. 2b). Canonical Wnt/β-catenin signaling is inhibited by dickkopf proteins (Dkk1–4), and secreted frizzled-related proteins (Sfrp1–5). In this context, *Dkk3* is highly expressed in the SVZ while *Sfrp1* and *Sfrp5* displayed higher levels in the choroid plexus (Fig. 2c). Interestingly, recent studies have reported that *Dkk3* is expressed in qNSCs of the SVZ[5]. Finally, we analyzed expression levels of Wnt co-receptors

*Lrp5, Lrp6, Ror2,* and *Ryk* in sorted *GFAP::GFP*[+] NSCs (Supplementary Fig. 1b), as these receptors have been reported to be crucial for inducing the canonical and non-canonical Wnt responses[35,14] (Fig. 2d, e). Interestingly, only *Ror2* transcripts were significantly upregulated in NSCs in the SVZ (Fig. 2e).

We next utilized a SVZ ex vivo culture system, which preserves both quiescent and activated NSCs as observed in vivo, to determine if the SVZ responds to canonical and non-canonical

Wnt signals (Fig. 2f). SVZ tissue was dissected from coronal brain sections and cultured on transwell inserts for 24 h before stimulation with either canonical Wnt3a or non-canonical Wnt5a conditioned media (see Methods and Supplementary Fig. 1c). Protein analysis showed that Wnt3a induced Axin2 expression (a canonical Wnt/β-catenin target), while Wnt5a reduced it. Interestingly, in the presence of both ligands, Wnt5a antagonized the Wnt3a-induced Axin2 levels (Fig. 2g, h). This experiment mimics the SVZ in vivo biological state where both canonical and non-canonical Wnt signals are expressed, but non-canonical Wnt signaling takes precedence by antagonizing β-catenin activity. Indeed, previous reports using transgenic reporter mice suggested that canonical Wnt/β-catenin activity is virtually absent in the SVZ during homeostatic conditions[26].

**Non-canonical Wnt induces activation of Rho-GTPase Cdc42.** Non-canonical Wnt signaling regulates many elements involved in the planar cell polarity pathway such as vangl2 and celsr2[15,25,36–38]. Furthermore, we also observed changes in the protein expression of Rho-GTPase Cdc42 effectors in our proteomics screen. Wnt5a-induced activation of Rho-GTPase Cdc42 is also reported as a key element in the maintenance of hematopoietic stem cell identity during physiology and aging[23,39]. During cortical development, Cdc42 shuttles between an active GTP-bound state and an inactive GDP-bound state, and this signaling is crucial in maintaining apical cell polarity and apical adhesion of radial glia[40]. We therefore asked whether Cdc42 activation is a downstream effector of non-canonical Wnt signaling in the adult SVZ to promote NSC polarity and adhesion to the apical niche. Using the ex vivo culture system described previously (Fig. 2f), we stimulated SVZ explants with Wnt5a and observed a marked enhancement in Cdc42-GTP levels as determined by a Cdc42-effector domain pull-down assay. This Cdc42 activation was inhibited in the presence of CASIN (Fig. 3a, b; CASIN is a Pirl1-related compound 2, a small-molecule inhibitor of Cdc42 activity referred in ref.[39]). Furthermore, we also analyzed Cdc42 expression in primary SVZ NSC cultures stimulated with Wnt5a in the presence and absence of CASIN (Fig. 3c). Interestingly, Wnt5a promoted NSC polarization as detected by a strong localization pattern of Cdc42 along with tubulin (Fig. 3c, indicated by arrows). Fluorescence intensity analysis of Cdc42 and tubulin across the NSC (as shown in the graphic representation) demonstrated that Wnt5a promotes NSC polarization (Fig. 3c, bottom panels). Conversely, treatment of NSCs with Wnt5a along with CASIN prevented this polarization, as NSCs in this condition displayed a rounded morphology (Fig. 3c, indicated by arrowheads). It was recently shown that N-Cadherin regulates apical NSC adhesion to the ventricular wall[9]. Given the role of Cdc42 and N-Cadherin in maintenance of polarity and cell adhesion[40,41], we asked whether Wnt5a regulates N-cadherin expression. As expected, Wnt5a elevated N-Cadherin levels in NSCs and this effect was abolished in the presence of CASIN (Fig. 3d–f). Altogether, our data indicate that non-canonical Wnt signaling-induced Cdc42 activity regulates SVZ NSC polarity and adhesion.

**Cdc42 signaling activity maintains SVZ NSC identity.** Given that non-canonical Wnt signaling plays a crucial role in regulating Cdc42 signaling activity, and thereby the adhesion of NSCs to the SVZ niche, we asked if this pathway also plays a role in regulating NSC identity. To this end, we stimulated SVZ explants with Wnt5a and analyzed the expression of molecules involved in stemness. Interestingly, Wnt5a induced expression of *Notch1*, *Hes5*, *Id1, and Id3*, and this occurred in a Cdc42-dependent manner (Fig. 3g). It should be noted that Wnt5a only

upregulated Notch1 receptor expression, but not its ligands Dll1 or Jag1 (Supplementary Fig. 2a, b). Given the role of Id1 and N-cadherin in maintaining the adhesion and functional quiescence of SVZ NSCs[9,12,42], and the role of Notch1 signaling in maintaining self-renewal[6], we hypothesized that activation of Cdc42 by non-canonical Wnt signaling regulates the quiescent state of NSCs, and hence disrupting Cdc42 activity in the SVZ will disrupt the quiescent state. To test this hypothesis in vivo, we first injected mice with CASIN (see Methods) and assessed Cdc42-GTP levels in SVZ lysates and noticed a sharp decrease compared to control (CTRL; vehicle injected) mice as early as 6 h after CASIN injection (Fig. 3h, i). We also observed reduced expression levels of Notch1, Id1, and N-cadherin in SVZ at 3 days after treatment with CASIN (Fig. 3j, k). These results suggest that systemic administration of CASIN results in transient inactivation of Cdc42 signaling, and consequently, decreased expression of its downstream targets.

In HSCs, increased Cdc42 activation was associated with aging-induced quiescence[23]. No other cytokines or growth factors have been shown to induce Cdc42 activation in adult NSCs to the best of our knowledge, and given that the major signaling mechanisms are conserved between NSCs and HSCs, we propose that non-canonical Wnt signaling-induced Cdc42 activation is critical in regulating the quiescent state of NSCs. Indeed, recent RNA sequencing studies by both Codega et al. [5] and Beckervorder-sandforth et al. [21] showed that quiescent NSCs express high levels of non-canonical Wnt/PCP pathway components that are required for Cdc42 activation. Hence, we asked whether Cdc42 activation is crucial to maintain the quiescent state of NSCs. To test this hypothesis functionally, we performed a BrdU label retention study[43]. Slowly dividing qNSCs retain the thymidine analog 5-bromodeoxyuridine (BrdU) label for long periods after administration as they remain quiescent, but when they become activated and undergo cell division they dilute away the BrdU label (Fig. 3l). We hypothesized that blocking Cdc42 activity will induce activation of qNSCs in the SVZ and hence they would dilute away the BrdU label. We first confirmed that BrdU label-retaining cells in the SVZ were indeed NSCs using Sox2 and GFAP antibodies (Fig. 3m). Furthermore, we detected fewer numbers of label-retaining quiescent NSCs in the SVZ after CASIN treatment (Fig. 3m, n). In addition to this readout, immunohistochemical analysis of ki67 revealed an increase in proliferating cells in the SVZ after CASIN treatment (Supplementary Fig. 2c, d). We also confirmed that the decrease in BrdU label-retaining cells (BrdU LRCs) was not due to cell death induced by CASIN treatment. Flow cytometric analysis of SVZ cells obtained 3 days after CASIN treatment labeled with annexin V and propidium iodide showed no significant changes in the numbers of apoptotic cells as compared to CTRL-treated mice (Supplementary Fig. 2e). These findings demonstrate that inhibiting Cdc42 activity disrupted the quiescent state of NSCs and triggered their activation. It must also be noted here that canonical Wnt/β-catenin signaling had no impact on Notch1 expression or its signaling effectors, as we did not see any alterations in protein levels of these elements in SVZ explants cultured with Wnt3a and Wnt3a+XAV939 (XAV939 is a tankyrase inhibitor that antagonizes canonical Wnt/β-catenin signaling; Supplementary Fig. 3a, b). However, we observed an upregulation in PCNA expression in SVZ explants upon stimulation with Wnt3a, indicative of increased proliferation (Supplementary Fig. 3c). To explore these findings further, we infected the SVZ of *GFAP::GFP* reporter mice with Wnt3a adeno-associated virus (Wnt3a-AAV) and analyzed brain sections from these mice 7 days later. Although we noticed an increase in ki67+ cells, we did not detect any alterations in apical *GFAP::GFP* cell numbers which are generally considered to be qNSCs

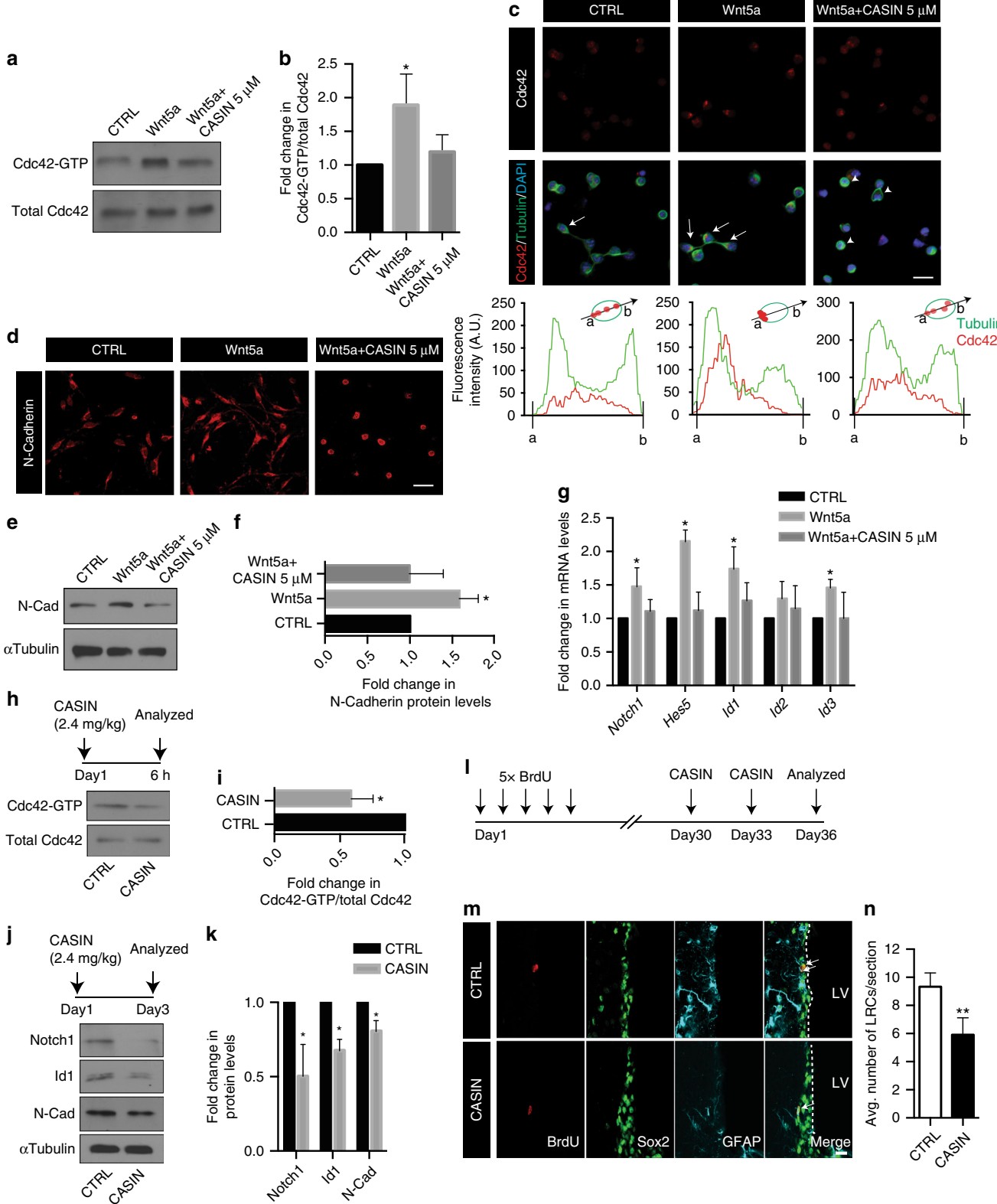

(Supplementary Fig. 3d–f). These results further corroborate that SVZ NSCs rely predominantly on non-canonical Wnt signaling to maintain their quiescent status.

**Demyelination induces alterations in SVZ Wnt signaling.** Our data suggest a physiological balance between non-canonical and

canonical Wnt signaling pathways in maintaining NSC cell cycle status. Further, given that our proteomics analysis showed that cell adhesion and Wnt signaling elements were altered in the SVZ at the peak of demyelination, we next explored the role of these pathways in quiescent NSC activation during injury. It must be noted that the acute demyelination model was used for these studies, as it has well-defined time points of early onset (3 days

**Fig. 3** Non-canonical Wnt signaling maintains NSC quiescence through the activation of Rho-GTPase Cdc42. **a** Wnt5a induces activation of Cdc42 in SVZ explant cultures as detected in Cdc42-GTP immunoblots, and this activation is inhibited by CASIN. **b** Densitometry quantification of **a**. Cdc42-GTP/total Cdc42 for Wnt5a and Wnt5a+CASIN 5 µM are represented as fold change over the control condition. **c** Cdc42 and tubulin staining in NSCs stimulated with control, Wnt5a, or Wnt5a+CASIN 5 µM. Bottom most panels show fluorescence intensity distributions of tubulin (green) and Cdc42 (red) along a single NSC as shown in the cartoon. Note the polarized pattern of Cdc42 and tubulin on one end of the cell in Wnt5a-treated cells. Scale bar represents 15 µm. **d** Wnt5a induces polarization and radial glial-like morphology in NSCs as depicted by N-Cadherin immunostaining. Scale bar represents 20 µm. **e** Wnt5a induces N-Cadherin protein expression in SVZ explants in a Cdc42-dependent manner. **f** Densitometry quantification of immunoblot shown in **e**. **g** Transcript levels of *Notch1*, *Hes5*, *Id1*, *Id2*, and *Id3* in SVZ explants after treatment with Wnt5a and Wnt5a+CASIN 5 µM represented as fold change over control. **h** Cdc42-GTP levels are decreased in SVZ protein lysates at 6 h following intraperitoneal injection of CASIN. **i** Densitometry quantification of Cdc42-GTP immunoblot shown in **h**. **j** CASIN treatment leads to decreased expression of Notch1, Id1, and N-Cadherin protein levels in the SVZ as compared to CTRL, after 72 h. **k** Densitometry quantification of immunoblots shown in **j**. **l** Experimental paradigm and timeline for detection of long-term BrdU label-retaining quiescent NSCs after CASIN treatment. **m** Inhibition of Cdc42 activity results in reduced numbers of qNSCs as detected by the numbers of BrdU label-retaining NSCs (BrdU LRCs, indicated by arrows). Scale bar represents 20 µm. **n** Quantification of BrdU LRCs in the SVZ of CTRL and CASIN-treated mice (total number of cells counted for CTRL: 298; CASIN: 189 cells). *$P < 0.05$, **$p < 0.01$; error bars represent mean ± s.e.m.; $n = 4$ for **b**, $n = 3$ for **f**, **g**, **i**, **k**; $n = 4$ for **n**; $n =$ independent experiments

post injection (3DPI)), peak of injury (7DPI), and myelin regeneration starting from 14DPI. To understand the alterations in Wnt signaling during injury, we first analyzed the mRNA levels of Wnt ligands that were enriched in the SVZ (Fig. 2a, b). Interestingly, the non-canonical ligand *Wnt4* was downregulated at 3DPI (Fig. 4a). By 7DPI, while *Wnt4* expression returned to basal levels, non-canonical *Wnt11* was downregulated significantly (Fig. 4a). In line with this expression pattern, we also detected a significant decrease in Cdc42 activity in the SVZ as revealed by reduced levels of Cdc42-GTP at 7DPI (Fig. 4b, c). We next analyzed the canonical Wnt ligands that were enriched in the SVZ (Fig. 2a) and observed that *Wnt7b* displayed a significant increase in expression from 3DPI to 7DPI, along with an increase in levels of its downstream effector active-β-catenin (Fig. 4d–f). In line with this finding, we also observed a significant increase in mRNA levels of the canonical Wnt targets *Axin2* and *Lef1* at 7DPI (Fig. 4g). These data suggest that a downregulation in non-canonical Wnt signaling is required for the activation of canonical β-catenin signaling in the SVZ during a demyelination injury.

To appreciate if the decrease in non-canonical Wnt elements and upregulation of β-catenin-dependent Wnt signaling during demyelination had any impact on the quiescent state of NSCs in the SVZ, we performed a BrdU label retention assay along with an acute demyelination injury (see Methods and Fig. 4h)[43,44]. Interestingly, we observed a reduction in the numbers of BrdU LRCs at the peak of demyelination (7DPI) as compared to control SVZ (Fig. 4i, j). To further corroborate that the decline in these LRC numbers was indeed due to their activation and cell cycle entry, we performed immunohistochemistry analysis using the S-phase proliferation marker proliferating cell nuclear antigen (PCNA). While in the contralateral (control; NaCl-injected hemisphere) SVZ BrdU LRCs did not express PCNA (Fig. 5a), we detected an increased number of diluted BrdU cells co-expressing PCNA (BrdU⁺PCNA⁺ cells; diluted BrdU⁺ cells have decreased BrdU intensity; Fig. 5b, c, indicated by arrowheads) in the ipsilateral SVZ (lysolecithin-injected hemisphere). We also noted that these diluted BrdU⁺PCNA⁺ cells appeared in clusters and were positioned away from the ventricular wall in dorsal, lateral, and ventral SVZ regions after injury (Fig. 5a, b, top, middle, and bottom panels, respectively). We believe these diluted BrdU⁺PCNA⁺ cells correspond to the aNSCs and their progeny, as many of these diluted BrdU⁺ cells were observed en route to corpus callosum lesions, where they co-expressed Olig2 (Supplementary Fig. 4a–c). Altogether, these results suggest that the shift in non-canonical to canonical Wnt signaling activates quiescent NSCs during a demyelination injury. These results are also in line with previous reports that suggest canonical Wnt signaling promotes oligodendrogenesis in the SVZ[45].

**The Wnt–Notch axis regulates lineage progression of NSCs.** Given that a decrease in Cdc42 activity led to a decrease in BrdU label retention capacity in NSCs, we further analyzed the SVZ niche using wholemount preparations from *GFAP::GFP* reporter mice to understand if these changes were associated with any cytoarchitectural remodeling during a demyelination injury. At 3DPI, the apical *GFAP::GFP⁺* cell numbers from various anterior–ventral and posterior–dorsal regions were unaltered (Fig. 5d, e). However, at 7DPI the density of these apical *GFAP::GFP⁺* cells was significantly decreased (Fig. 5d, e, indicated by arrows). We next asked if this reduction in *GFAP::GFP⁺* cell numbers during demyelination was due to a decrease in apical adhesion marker expression and lineage progression. To this end, we first analyzed the expression pattern of molecular markers that promote apical NSC adhesion to the ventricular wall and ependymal cells that line the ventricular wall such as JAM-C[46], E-Cadherin[47], and N-Cadherin[9]. At 3DPI, JAM-C, E-Cadherin, and N-Cadherin expression levels in apical GFAP⁺ NSCs were markedly reduced (Supplementary Fig. 5a–c). We also observed a decrease in JAM-C levels in our proteomics screen (Supplementary Data 1). Interestingly, we additionally detected that planar cell polarity pathway elements *celsr2* and *vangl2* showed a reduction in transcript levels at the peak of demyelination (Supplementary Fig. 5d). Immunohistochemical analysis of Id1, another molecular marker of qNSCs[42] and an important regulator of NSC adhesion to the niche[12], showed decreased co-localization with *GFAP::GFP⁺* NSCs at this time point (Supplementary Fig. 5e). These data suggest that a decrease in Cdc42 activity during an acute demyelination injury leads to reduced levels of adhesion molecules that maintain NSC anchorage to the apical SVZ niche.

Given that non-canonical Wnt signaling induces Notch signaling activity, we also analyzed the *Hes1* and *Hes5* mRNA levels in the SVZ during a demyelination injury. Consistent with our data that showed decreased non-canonical Wnt signaling at the peak of injury, *Hes1* and *Hes5* mRNA levels were sharply downregulated at 7DPI (Supplementary Fig. 5f). To test if this decrease in Notch signaling had a direct impact on the self-renewal capacity of NSCs, we sorted *GFAP::GFP⁺* cells from contralateral and ipsilateral SVZ at 7DPI and performed in vitro neurosphere formation assays. Neural stem cells obtained from the ipsilateral SVZ at 7DPI formed fewer neurospheres (Fig. 5f, g). Supporting these data further, we also observed increased lineage progression as detected by increased numbers of GFAP⁻EGFR⁺ progenitor cells in the ipsilateral SVZ[48] (Fig. 5h, i). These data further confirm that non-canonical Wnt signaling, by regulating Notch, plays a major role in regulating NSC identity and their properties.

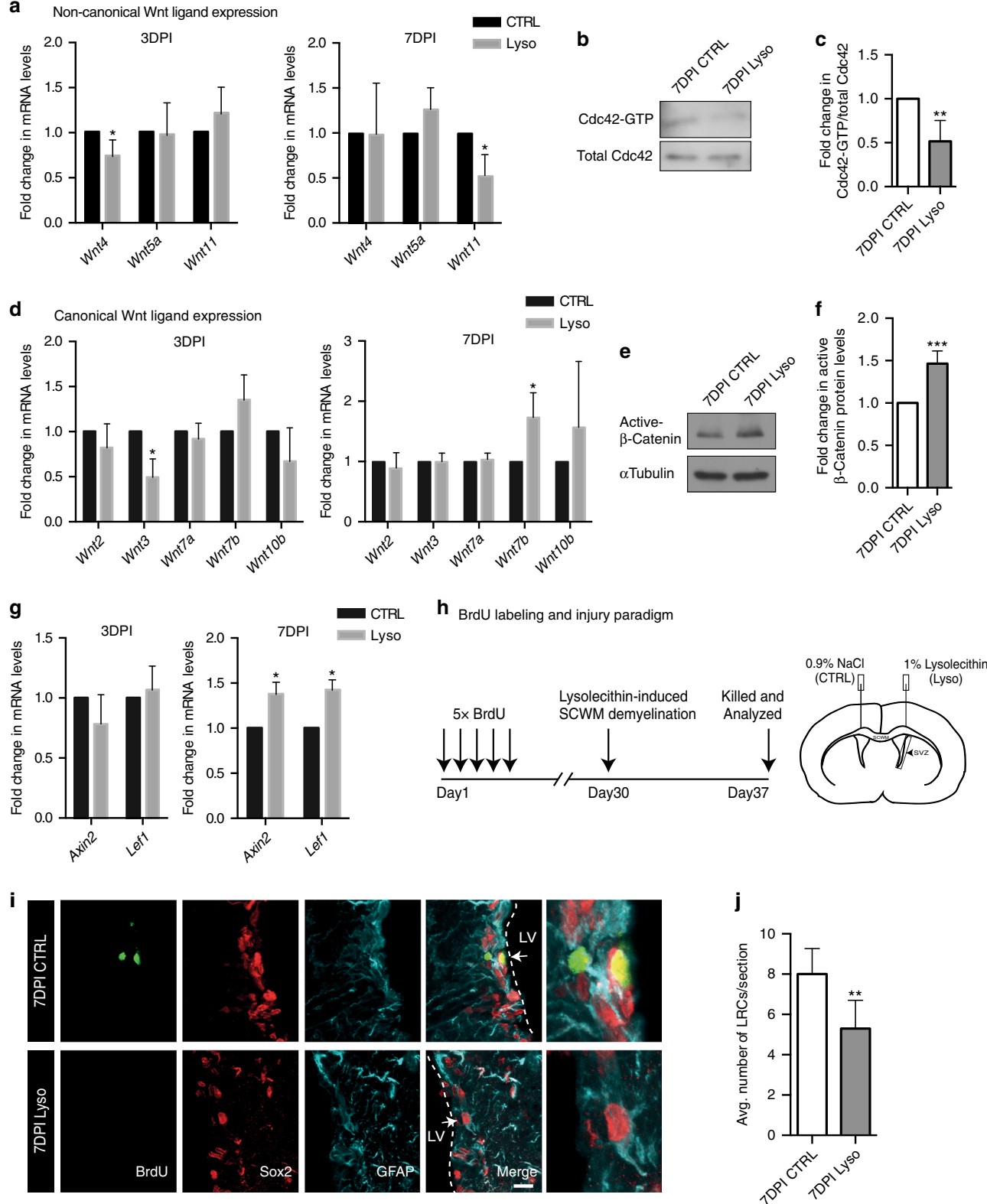

**Non-canonical Wnt preserves functional quiescence in the SVZ**. As our results indicated that a disruption in the balance between non-canonical and canonical Wnt signaling promotes activation of quiescent NSCs, we asked if we could halt this process by force inducing non-canonical Wnt signaling in the SVZ during demyelination. To this end, we used a gain-of-function approach using non-canonical Wnt (Wnt5a) or control

vector (LacZ) AAVs in the SVZ during demyelination. We first confirmed the ability of Wnt5a to induce non-canonical Wnt signaling by assessing nuclear translocation of β-catenin in AAV-transduced primary NSCs (see Methods). In the NSCs infected with Wnt5a, virtually no nuclear β-catenin localization was detected as compared to the control vector condition (LacZ; Supplementary Fig. 6a; top and middle panels), validating the

**Fig. 4** A transient shift from non-canonical to canonical Wnt Signaling in the SVZ during demyelination induces activation of quiescent NSCs. **a** Transcript levels of non-canonical Wnt ligands in the SVZ at 3DPI and 7DPI. Transcript levels of each gene in demyelination samples (lysolecithin injected/ipsilateral side) are represented as fold change over the control SVZ (NaCl injected/contralateral side) at the corresponding time point. **b** Immunoblot showing a decrease in Cdc42-GTP levels in the SVZ at the peak of demyelination (7DPI). **c** Densitometry quantification of **b**. Cdc42-GTP/total Cdc42 in the SVZ at 7DPI is represented as fold change over the control SVZ. **d** Transcript levels of canonical Wnt ligands in the SVZ at 3DPI and 7DPI. Transcript levels of each gene in demyelination samples are represented as fold change over the control SVZ at the corresponding time point. **e** Immunoblot showing an increase in active β-catenin (Non-phospho Ser33/37/Thr41 β-Catenin) levels in the SVZ at 7DPI. **f** Densitometry quantification of active β-catenin levels in the SVZ at 7DPI is represented as fold change over the control SVZ. **g** Transcript levels of *Axin2* and *Lef1* in the SVZ at 3DPI and 7DPI. Transcript levels of each gene at 3DPI and 7DPI are represented as fold change over the control SVZ at the corresponding time point. **h** Timeline and experimental paradigm to detect long-term BrdU label-retaining quiescent NSCs after induction of focal demyelination. **i** Representative images of BrdU label-retaining quiescent NSCs in the SVZ at 7DPI (peak of demyelination; indicated by arrows in top panel). Rightmost panels show higher magnification images of the indicated (arrows) cells. Scale bar represents 20 μm. **j** A decrease in the number of BrdU label-retaining quiescent NSCs in the SVZ at 7DPI was observed (total number of cells counted for CTRL: 264; Lyso: 197). *P < 0.05, **p < 0.01, ***p < 0.001; error bars represent mean ± s.e.m.; n = 4 for **a**, **c**, **d**, **f**, **g**; n = 6 for **j**; n = independent experiments

ability of Wnt5a-AAV to activate non-canonical Wnt signaling. Note that a high number of NSCs with nuclear β-catenin were observed after infection with canonical Wnt AAV (Wnt3a; Supplementary Fig. 6a; bottom panel), indicative of active canonical Wnt signaling.

To test the role of non-canonical Wnt signaling in maintaining NSC quiescence in vivo, we analyzed BrdU label retention capacity of NSCs during demyelination after force inducing Wnt5a expression in the SVZ (Fig. 6a, b; also see Methods). We first confirmed the targeting and infection efficiency of Wnt-AAVs in SVZ NSCs (Supplementary Fig. 6b). At 7DPI, a reduced number of GFAP+ cells co-expressed Axin2 in Wnt5a-AAV-infected SVZ as compared to LacZ- or Wnt3a-AAV-infected SVZs (Supplementary Fig. 6c–e). In addition, we also noted a decrease in *Axin2* mRNA levels in the SVZ after infection with Wnt5a and an increase after infection with Wnt3a (Supplementary Fig. 6f, g), confirming efficient transduction of the SVZ with Wnt vectors. Next, we proceeded to analyze the label retention capacity of NSCs in this experimental setting (Fig. 6a, b). Interestingly, while at 7DPI a reduced number of BrdU LRCs were observed in LacZ-AAV-infected SVZ (Fig. 6c, d), Wnt5a-AAV-infected SVZ did not have any significant differences in numbers of BrdU LRCs as compared to the control SVZ (Fig. 6c, d). To further analyze NSC activation status after injury we quantified ki67+ cell numbers in the SVZ at 7DPI after LacZ or Wnt5a-AAV infection. An increased number of ki67+ cells were evident in LacZ-infected SVZ, but this expansion was hindered by Wnt5a (Fig. 6e, f). These results suggest that Wnt5a-induced non-canonical Wnt signaling is crucial for the maintenance of NSC quiescence in the SVZ. It must be noted that Wnt3a could not rescue the decline in qNSC numbers as indicated by a decrease in BrdU LRC numbers in Wnt3a-AAV-infected SVZ during demyelination (Supplementary Fig. 6h). We believe this could be due to the inability of Wnt3a to affect the qNSC population as demonstrated in Supplementary Fig. 3.

We next asked if the Wnt5a-mediated rescue of BrdU LRC numbers is indeed due to its ability to induce Cdc42 activation. To this end, we force induced Wnt5a expression in the SVZ during a demyelination injury as described above (Fig. 6a, b) along with a transient loss of Cdc42 function achieved by systemic injection of CASIN (as described in Fig. 3). Intriguingly, Wnt5a's ability to prevent the activation of quiescent NSCs was lost when Cdc42 activity was inhibited by CASIN, as we observed reduced numbers of BrdU LRCs and an increase in ki67+ cells in the SVZ at the peak of demyelination (Fig. 6g–i). It must also be noted that the number of LRCs in the control SVZ was reduced due to systemic administration of CASIN, as described in Fig. 3. These results further demonstrate that a balance between non-canonical and canonical Wnt signaling tightly regulates NSC

quiescence in the SVZ in homeostatic conditions, and disruption of this balance during injury triggers the activation of quiescent NSCs in a Cdc42-dependent manner.

**Non-canonical Wnt activity in the SVZ delays remyelination.** We next wanted to examine if the decrease in non-canonical Wnt signaling in the SVZ is crucial for subcortical white matter (SCWM) remyelination after a demyelination injury. To this end, we force induced either LacZ or Wnt5a expression in the SVZ using AAVs during demyelination (as described earlier) and assessed remyelination at 14DPI by electron microscopy. Interestingly, we found fewer remyelinated axons in the SCWM of mice infected with Wnt5a-AAV compared to LacZ-AAV (Fig. 7a, b). Given that non-canonical Wnt signaling regulates Notch activity in NSCs in our system (Fig. 3g), we also wanted to determine how overactivation of Notch signaling in the SVZ during demyelination affected the remyelination process. To this end, we employed a gain-of-function approach using activated-Notch-AAV (NICD-AAV) and infected the SVZ during demyelination (Supplementary Fig. 7a, b). We first confirmed the infection efficiency and upregulation of NICD/*Hes5* levels in SVZ NSCs (Supplementary Fig. 7c–f). We then assessed SCWM remyelination at 14DPI by electron microscopy. Intriguingly, we observed a severe block in SCWM remyelination when NICD was overexpressed in the SVZ, as the number of remyelinated axons was significantly reduced (Fig. 7a, b). The degree of remyelination appeared to be far more hindered when NICD was overexpressed in the SVZ as compared to Wnt5a (an average of 46 remyelinated axons/field in the Wnt5a condition as compared to 37 axons/field in the NICD condition). We believe this could be due to a potent differentiation block that NICD can induce (given its ability to function as a transcription co-factor in the nucleus) on the NSCs and also NPCs that are further along in lineage progression and already present in the SVZ at the time of injury onset, while overexpression of Wnt5a might only affect the quiescent NSCs that would have normally undergone activation after demyelination. To confirm if NICD-infected mice recovered at a later point, we examined the functional recovery in our demyelination system by registering extracellular compound action potentials (CAPs), a method used to assess the presence of functional myelin and remyelination[49]. CAPs are field potentials recorded in the SCWM that represent the signal propagating through the fast-conducting M (myelinated) and the slower conducting UM (unmyelinated) fibers. CAPs show two distinct negative waves with different conduction velocities (Supplementary Fig. 7g, top panel). To determine the rate of remyelination, CAPs were recorded at 7DPI and 18DPI. At 7DPI, CAP signals showed complete loss of the M wave in both AAV-LacZ and AAV-NICD

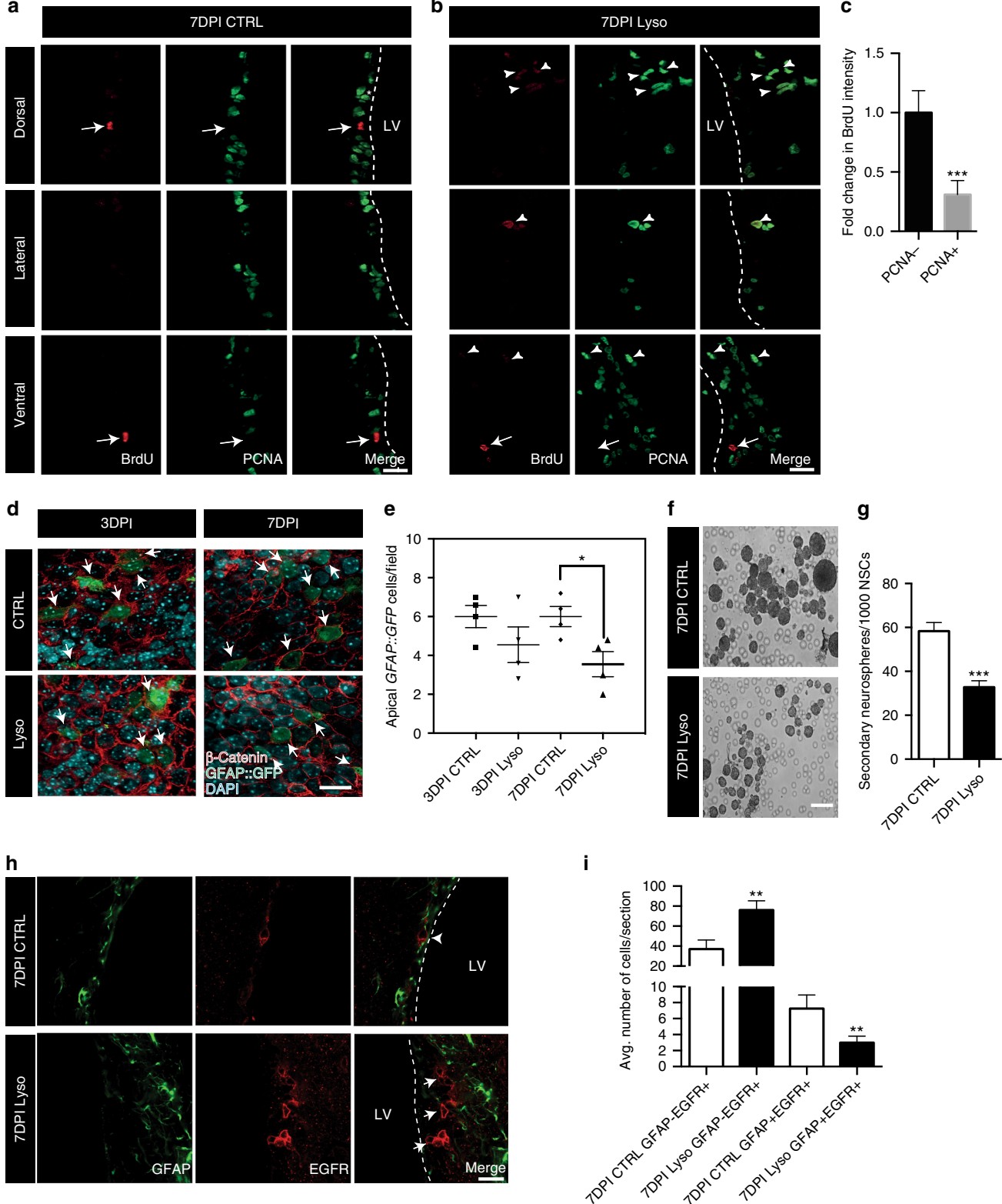

conditions (Supplementary Fig. 7g, h, middle panel). No changes in amplitude and conduction velocity of the UM wave were detected at 7DPI, confirming the degree of demyelination was similar in both the LacZ- and NICD-transduced animals (Amplitude: control, $0.23 \pm 0.04$ mV, $n = 19$; LacZ: $0.25 \pm 0.01$ mV, $n = 16$; NICD: $0.25 \pm 0.04$ mV, $n = 17$; $p = 0.83$, analysis of variance (ANOVA); Conduction velocity: control,

$0.30 \pm 0.01$ ms$^{-1}$, $n = 19$; LacZ: $0.30 \pm 0.01$ ms$^{-1}$, $n = 16$; NICD $0.30 \pm 0.01$ ms$^{-1}$, $n = 17$; $p = 0.7$, ANOVA). However, at 18DPI, a distinct peak with conduction velocity corresponding to the M wave could be detected in LacZ animals (Supplementary Fig. 7g, h, bottom panel). No recovery of the M component of the electrophysiological signal was detected in NICD-AAV-transfected animals (Supplementary Fig. 7g, bottom panel; $0.76 \pm 0.03$ ms$^{-1}$,

**Fig. 5** Increased lineage progression in SVZ NSCs during demyelination. **a, b** Representative images of BrdU label-retaining quiescent NSCs and their activated progeny (BrdU⁺PCNA⁺ cells) in the **a** contralateral and **b** ipsilateral SVZ at dorsal, lateral, and ventral levels (top, middle, and bottom panels, respectively) at 7DPI. The absence of PCNA reactivity within the BrdU label-retaining cell population is indicative of quiescent NSCs (indicated by arrows) and its expression within the diluted BrdU⁺ population denotes mitotically active cells (indicated by arrowheads). Note that most BrdU⁺ label-retaining NSCs in the control SVZ are localized to the ventricular wall (**a** all panels), while after demyelination diluted BrdU⁺PCNA⁺ cells can be observed in the periventricular region (**b** all panels). Scale bar represents 25 μm. **c** Quantification of BrdU intensity of both PCNA⁻ (quiescent) and PCNA⁺ (activated) cells represented as a fold change. Note that PCNA⁻ quiescent LRCs have a strong BrdU label, while the PCNA⁺ activated cells have decreased BrdU intensity as they dilute away the label while dividing. **d** Representative images of SVZ wholemounts (en-face view) showing apical SVZ NSCs (*GFAP::GFP*⁺ cells) intercalated in the ependymal layer delineated by β-catenin staining at 3DPI and 7DPI. Note that the number of apical *GFAP::GFP*⁺ cells is decreased at peak of demyelination (7DPI; indicated by arrows). Scale bar represents 20 μm. **e** Quantification of apical *GFAP::GFP*⁺ cell numbers at 3DPI and 7DPI. **f** Neurospheres derived from the contralateral (control hemisphere) and ipsilateral (demyelinated hemisphere) SVZ NSCs at 7DPI. Scale bar represents 50 μm. **g** Decrease in secondary neurosphere numbers from ipsilateral SVZ NSCs compared to contralateral. **h, i** Characterization of NSC lineage progression in the contralateral and ipsilateral SVZ at 7DPI. Demyelination decreases GFAP⁺EGFR⁺ activated NSC density (indicated by arrowheads) and increases the pool of GFAP⁻EGFR⁺ progenitors in the SVZ (indicated by arrows). Scale bar represents 50 μm. *$P < 0.05$, **$p < 0.01$, ***$p < 0.001$; error bars represent mean ± s.e.m.; $n = 3$ for **c**, $n = 4$ for **e**, $n = 3$ for **g**, $n = 4$ for **i**; $n =$ independent experiments

$n = 19$; LacZ $0.69 \pm 0.03$ ms⁻¹, $n = 11$; $p = 0.09$, ANOVA). Altogether, our results suggest that Notch signaling in the SVZ is sufficient to maintain NSC self-renewal and identity, and that a decrease in its activity in a non-canonical Wnt-dependent manner during a demyelination injury ensures lineage progression of NSCs and remyelination.

## Discussion

In the adult brain, the unique SVZ niche microenvironment provides NSCs access to contact-mediated signals within the SVZ, signaling cues emanating from the cerebrospinal fluid secreted by choroid plexus[50], and angiocrine signals that arise from the vascular network. These signals are critical for the maintenance of functional quiescence and activation of NSCs[11,51–54]. Our study demonstrates that quiescence is a dynamically maintained state in the SVZ and we identified non-canonical Wnt signaling as a key pathway in regulating this process. Further, we uncovered that qNSCs are recruited into the cell cycle by demyelination injury-induced alterations to non-canonical Wnt signaling and adhesion molecules in the SVZ, and these mechanisms are critical to generate actively dividing NSCs that undergo lineage progression to achieve tissue homeostasis and functional repair of the SCWM.

In the adult SVZ, there is a well-defined relationship between NSC positioning within the niche and cell cycle status[7,9,10]. Luo et al. [7] demonstrated that NSCs lining the ventricular wall are quiescent, while mitotically active NSCs reside in the periventricular region close to the vascular network. In this context, lineage progression of stem cells is a dynamic process that involves several cellular and molecular mechanisms along with spatial positioning within the niche. A classic example of this mechanism is the male germline stem cell niche of *Drosophila*, where stem cells divide asymmetrically, and one cell remains in the niche and maintains a stem cell identity, whereas the daughter cell is displaced out of the niche and undergoes differentiation[55]. Indeed, Porlan et al. [9] have recently proposed that a similar mechanism regulates NSC activation by post-translational regulation of N-cadherin. Our data further support this idea, and that along with positioning within the niche, additional spatial and temporally compartmentalized signaling cues regulate NSC mitotic status. We focused our work on Wnts as candidate signals that regulate NSC properties, as ligands for activation of this pathway are secreted by both the choroid plexus[28,56] and the SVZ niche cells, and NSCs have access to these signals via their primary cilium[5].

Our data reveal that non-canonical Wnt signaling-induced activation of Cdc42 is crucial in regulating the expression of adhesion molecules that keep the NSCs anchored to the SVZ niche and to promote expression of stemness inducing molecules

such as Notch1 and Id1[6,42]. We validated that SVZ cells respond to both canonical and non-canonical Wnt ligands, albeit in a different manner. Genetic reporter analysis revealed that β-catenin signaling activity is almost absent in SVZ NSCs in homeostatic conditions[26]. Our data here are in line with this work, as we noticed that even in the presence of both signals, non-canonical Wnt ligand takes precedence over the canonical Wnt ligand and suppresses β-catenin signaling. One explanation for this is that an increase in active Cdc42 levels could recruit β-catenin to interact with its binding partners such as α-catenin and N-cadherin. Indeed, we also observed a strong increase in N-cadherin levels after stimulation of SVZ explants with Wnt5a. Therefore, the levels of active Cdc42 could dictate whether β-catenin undergoes nuclear translocation to induce transcriptional activity or functions as a cell adhesion component. Although we did not characterize the expression of Wnt ligands in the choroid plexus following a demyelination injury due to technical limitations, we believe it will be an important source of differential Wnt expression following an injury, which might affect SVZ NSCs in this setting.

Importantly, we also show that this mechanism is crucial in regulating NSC quiescence during brain injury. Our data reveal that the activation of SVZ NSCs after a demyelination injury is dependent on downregulation of active Cdc42 levels which in turn downregulates Notch signaling and allows for an upregulation of Wnt/β-catenin targets. Canonical Wnt signaling has been reported to have a role in the generation of oligodendrocyte precursors[28,45]. As our data show an upregulation of Wnt/β-catenin signaling in the SVZ during a demyelination injury, it is tempting to suggest this shift in the balance between non-canonical and canonical Wnt signaling in the SVZ might be an adaptive process to changes in microenvironment post injury, as overriding this activation of β-catenin signaling by either Wnt5a or Notch gain of function resulted in failure to achieve remyelination.

Overall, our data support a model (Fig. 8a) of SVZ qNSC maintenance based on molecular properties regulated by Wnt pathway elements. Non-canonical Wnt signaling in qNSCs induces activation of Rho-GTPase Cdc42, which in turn regulates expression of Notch1, Id1, and N-Cadherin, thereby maintaining NSC self-renewal and anchorage to the SVZ niche. During a demyelination injury, downregulated non-canonical Wnt ligand expression in the SVZ leads to a transient decrease in Cdc42 activity and an increase in canonical Wnt/β-catenin signaling. This shift in non-canonical to canonical Wnt activity induces NSC activation and lineage progression to generate regenerative progenitors that eventually go on to participate in the repair process (Fig. 8b).

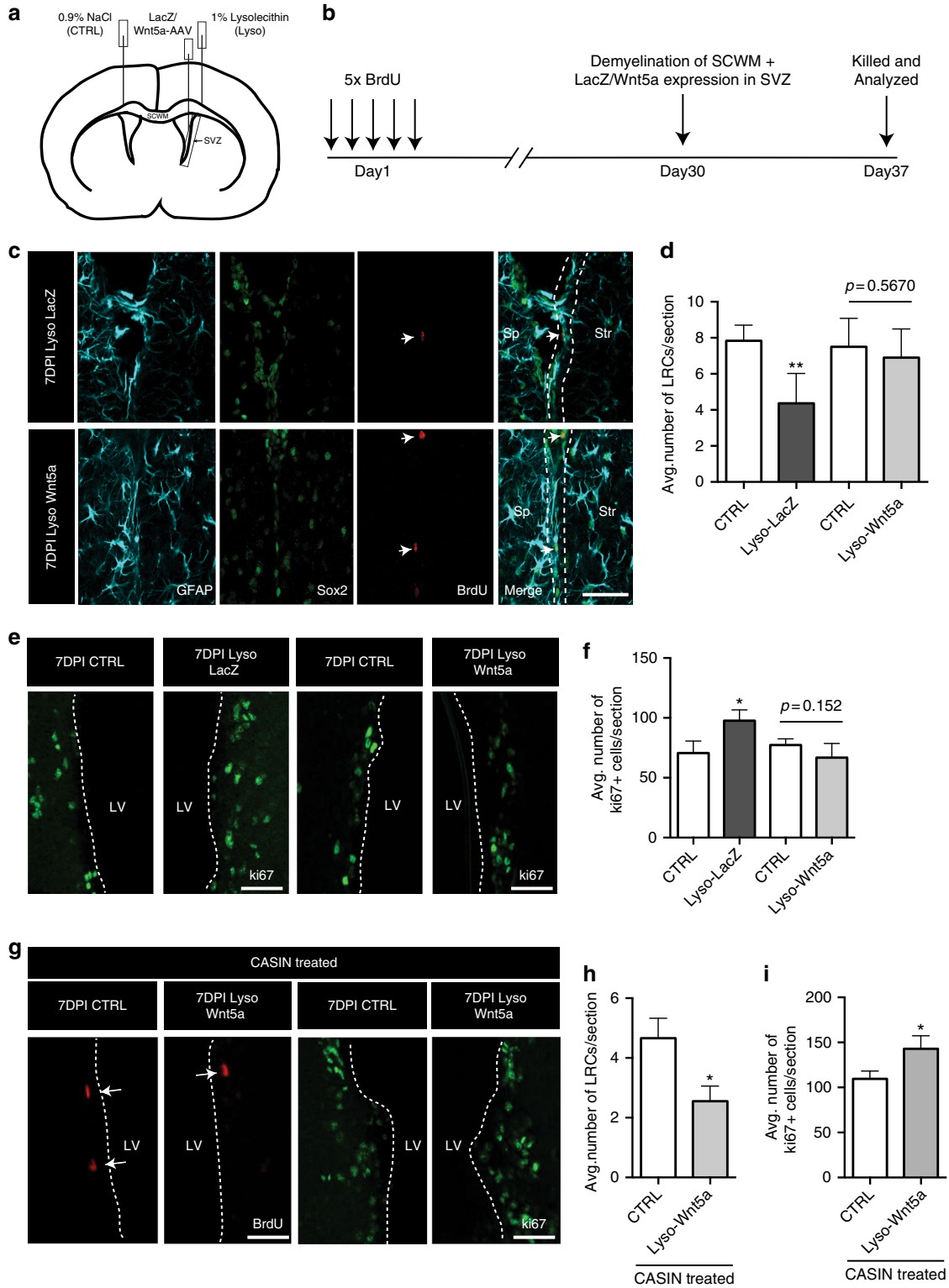

## Methods

**Animals**. All animal procedures are approved by the Institutional Animal Care and Use Committee, SUNY Stony Brook, School of Medicine and the National Institutes of Health "Guide for the Care and Use of Laboratory Animals." Adult C57BL/ 6 male and female (2–3 months old) mice were used in this study. Heterozygous *GFAP::GFP*[57] (FVB/N-Tg(GFAPGFP)14Mes/J, Jackson Labs strain 003257) mice were used wherever mentioned.

**Focal demyelination induced by lysolecithin**. C57BL/6 and *GFAP::GFP* (2–3 months old) mice were deeply anesthetized with a ketamine/xylaxine cocktail

(10 mg/ml ketamine; 1 mg/ml xylazine) and positioned on a stereotaxic frame (Stoelting). A 32-gauge, two-inch needle was attached to a 5 μl Hamilton syringe and mounted on a stereotaxic micromanipulator. The demyelinating agent (2 μl of 1% Lysolecithin in 0.9% NaCl) was injected unilaterally into the corpus callosum using stereotaxic coordinates of 1.5 mm anterior–posterior; 1 mm mediolateral and 2 mm dorsoventral from Bregma. The needle was kept in place for 5 min to reduce reflux along the needle track[49,58]. The day of injection was noted as day 0, and this served as the starting time point for all measurements of DPI. For this model, 3DPI served as the earliest time point of the injury and 7DPI served as the peak of demyelination, while 21DPI served as the recovery time point. The contralateral

**Fig. 6** Overexpression of non-canonical Wnt signaling maintains the quiescent state of NSCs in the SVZ during demyelination. **a, b** Experimental paradigm and analysis timeline of non-canonical Wnt signaling gain of function in the SVZ during demyelination. **c** Representative images of BrdU label-retaining quiescent NSCs from ipsilateral SVZ at 7DPI after LacZ and Wnt5a-AAV infection of the SVZ. Forced expression of Wnt5a in the SVZ prevents the demyelination-induced decline in BrdU label-retaining quiescent NSCs (indicated by arrows). Sp -septum, Str -striatum. Scale bar represents 50 μm. **d** Quantification of BrdU label-retaining quiescent NSCs at 7DPI after LacZ and Wnt5a-AAV infection of the SVZ (total number of cells counted for CTRL: 235; Lyso-LacZ:134; CTRL:225; Lyso-Wnt5a:207). **e** Representative images of ki67[+] cells in the SVZ at 7DPI after LacZ and Wnt5a-AAV infection. Wnt5a overexpression results in a decrease in ki67[+] cells in the SVZ during a demyelination injury. Scale bars in **e** represent 50 μm. **f** Quantification of ki67[+] cells in the SVZ at 7DPI after LacZ and Wnt5a-AAV infection during demyelination (total number of cells counted for CTRL: 636; Lyso-LacZ:865; CTRL:929; Lyso-Wnt5a:802). **g** CASIN inhibits Wnt5a-induced rescue of quiescent NSC activation. Left most panels show representative images of BrdU label-retaining quiescent NSCs at 7DPI after Wnt5a-AAV infection and CASIN treatment (indicated by arrows). Rightmost panels show ki67[+] cells in the SVZ under same experimental conditions. Scale bars represent 50 μm. **h** Quantification of BrdU label-retaining quiescent NSCs in the SVZ at 7DPI after Wnt5a infection and CASIN treatment (total number of cells counted for CTRL:70; Lyso-Wnt5a:35). **i** Quantification of ki67[+] cells in the SVZ at 7DPI after Wnt5a-AAV infection and CASIN treatment (total number of cells counted for CTRL:1199; Lyso-Wnt5a:1287). $*P < 0.05$, $**p < 0.01$; error bars represent mean ± s. e.m.; $n = 5$ for **d**, $n = 4$ for **f**, and $n = 3$ for **h, i**; $n =$ independent experiments

side of the same mice was used as a control unless noted otherwise in the Results and figure legends. The corpus collosum from the non-injured control hemisphere was injected with 1 μl of 0.9% sterile NaCl (referred to as CTRL).

**Proteomics analysis of the SVZ during demyelination**. Proteomics data were obtained from SVZ tissue of chronically demyelinated adult C57/Bl6 mice that was achieved by 0.2% Cuprizone (cyclohexylidenehydrazide; Sigma-Aldrich; fed ad libitum mixed in pelleted chow diet (Teklad)[58] treatment for 12 weeks. The GO and pathway enrichment analyses of proteomics data from SVZ tissue at the peak of demyelination were performed using the Database for Annotation, Visualization and Integrated Discovery (DAVID) software and Reactomics Pathway analysis tool followed by the construction of a protein–protein interaction network. Proteins upregulated by 1.2-fold and downregulated by 0.8-fold were used on the GO consortium website at geneontology.org to detect alterations in specific categories. The tandem mass spectrometry (MS/MS) data were first analyzed using a binary classifier that is trained on a manually validated data set to remove the low-quality MS/MS spectra. The resulting spectra were analyzed against a mouse protein FASTA sequence database EBI-IPI (European Bioinformatics Institute–International Protein Index) [59], and a decoy database with reversed sequences[60] was used to calculate the confidence levels and the false positive rates. The spectra were further validated by DTAselect[61]. Furthermore, a minimum of 7 amino acid residues sequence length was required, and two independent peptide identifications was required to reduce the false discovery rate to less than 1%. Finally, spectral counting methods[62,63] were used to obtain the fold changes.

**RNA isolation and quantitative PCR**. RNA from SVZ tissue or cells was isolated using trizol (Ambion) and chloroform. The total RNA was separated from the aqueous phase using MicroKit (QIAGEN). Complementary DNA (cDNA) was reverse transcribed using random primers and Superscript IV (Invitrogen). Quantitative PCR was performed from cDNA using Power Up SYBR Green master mix (Invitrogen) as per the manufacturer's protocol on an Applied Biosystems 7900HT real-time PCR system (Ct values above 35 cycles were not considered for analysis). Fold changes were calculated using the ΔΔCt method. For the primer sequences, see Supplementary Data 3. All genes of interest were normalized to *Gapdh* mRNA levels.

**Wnt conditioned medium**. Control (CTRL), Wnt3a, and Wnt5a conditioned medium was produced in L-cell lines[64,65] that were a kind gift from Dr. Joel Levine (Stony Brook University). L-cell lines were maintained in Dulbecco's Modified Eagle's medium (DMEM) with 10% fetal bovine serum (FBS) plus 1× antibiotics. When cells reached 70% confluence, maintenance medium was replaced with neural stem cell medium (1:1 DMEM/F12 medium, 1× B27 supplement, and 1% penicillin/streptomycin) with 1% FBS. Wnt proteins tend to precipitate in the absence of serum, and therefore we used neural stem cell medium supplemented with 1% FBS. After 48 h, the conditioned medium was collected and centrifuged at 1500 RPM for 5 min, filtered through a 0.2 μm pore filter, and used for the stimulation experiments.

**Luciferase assay**. A HEK293 cell line with a stably integrated firefly luciferase cassette under the control of a TCF/LEF responsive element was generated using CIGNAL lentiviral particles purchased from Qiagen. Cells were plated in 12-well dishes and when they reached 80% confluence, they were stimulated with either CTRL, Wnt3a, or Wnt5a conditioned medium. Luciferase activity was determined at 4 h after stimulation using the Dual Assay Luciferase kit (Promega). All luminescence measurements were obtained using the Turner 20/20n luminometer.

**SVZ tissue explant cultures**. Brains from adult mice were washed twice in Hanks balanced salt solution (HBSS) and transferred to ice-cold HBSS medium prior to SVZ dissection. The brains were then cut into 200 μm thick sections using the McIlwain tissue chopper, and only the sections containing the SVZ were selected for tissue culture[66]. SVZ explants were cultured on 0.4 μm polycarbonate transwell inserts and fed with neural stem cell medium (NSC media; 1:1 DMEM/F12 with 1× B27 and 1× antibiotics) supplemented with 10 ng/ml FGF2. After 24 h, NSC medium was replaced with CTRL, Wnt3a, or Wnt5a conditioned medium without any other additional growth factors for 16 h. At the end of this period, samples were processed for western blot or RNA analysis.

**Immunoblotting assay**. For western blots, SVZ tissue or explants were collected at the end of the respective experiments and protein was extracted using RIPA lysis buffer with a protease inhibitor cocktail (Santa Cruz Biotechnologies, sc-24948). Protein samples were cleared by centrifugation (10,000×g for 5 min) and protein concentration was determined using the BCA assay (Pierce). Samples (10–15 μg) were boiled with SDS loading buffer, separated on acrylamide gels, then transferred to polyvinylidene difluoride membranes (Millipore). Primary antibodies were used for detection of the indicated proteins in combination with secondary horseradish peroxidase-conjugated antibodies (Santa Cruz Biotechnologies; 1:5000) and an enhanced chemiluminescence substrate mixture (ECL Plus, GE Healthcare). Where indicated, protein levels are quantified by densitometry in arbitrary units (A.U.) following normalization to α-tubulin levels (used as the loading control). All quantifications were performed using ImageJ. Primary antibodies, dilutions, and catalog numbers are listed in Supplementary Data 3. The blots were cropped for figure presentation and full-length blots are presented in Supplementary Figs. 8 and 9.

**Cdc42-GTPase effector domain pull-down assays**. GTP-bound Cdc42 levels were assessed by effector domain pull-down assays using a Cytoskeleton CDC42 activation assay kit (#BK034) as per the manufacturer's protocol. Protein lysates were prepared as above and incubated with Pak-PBD affinity beads for 1–2 h at 4 °C. Immunocomplexes bound to the beads were collected by centrifugation, washed twice in 500 μl RIPA buffer containing inhibitors, and beads were boiled with SDS loading buffer. Immunoblotting was performed as described above, and Cdc42 levels were detected with a monoclonal Cdc42-specific antibody.

**Inhibition of Cdc42 activity**. Pirl1-related compound 2 (CASIN, Sigma)[67] was used to block Cdc42 activation in vivo and in vitro at a concentration of 2.4 mg/kg (referred to as CASIN) and 5 μM (referred to as CASIN 5 μM), respectively. To test the efficacy of CASIN on SVZ cells, mice received a single CASIN injection (2.4 mg/kg) and 6 h later SVZ tissue was isolated to determine active Cdc42-GTP levels. For label retention experiments using CASIN, mice were injected with BrdU (60 mg/kg) five times/3 h apart and then subjected to a 30-day chase period. At the end of the BrdU chase period, mice received CASIN injections (two times, 3 days apart at 2.4 mg/kg on day 30 and day 33) and animals were killed on day 36 to perform the analysis of BrdU LRCs in the SVZ.

**Fluorescence-activated cell sorting**. Adult SVZ tissue from each experimental condition was dissected and processed independently for fluorescence-activated cell sorting (FACS)[6]. SVZ tissue was collected in ice-cold D1 media (1× HBSS, 1× antibiotics, 26 mM HEPES, 0.3% glucose, and 0.75% sucrose). For tissue processing, dissected SVZ was incubated with 0.1% trypsin and 100 U/ml DNase I in D1 for 25 min at room temperature and single-cell dissociation was achieved using a fire-polished glass Pasteur pipette. The single-cell suspension was then analyzed for light forward and side scatter and height and width to exclude dead cells and doublets using the FACSAria instrument (Beckton Dickinson). Positive and negative cell populations were either used for total RNA extraction or cell culture experiments. For GFAP::GFP[+] cell sorting experiments, cells with the highest green fluorescent protein (GFP) expression were sorted and considered as NSCs[5,68]. For

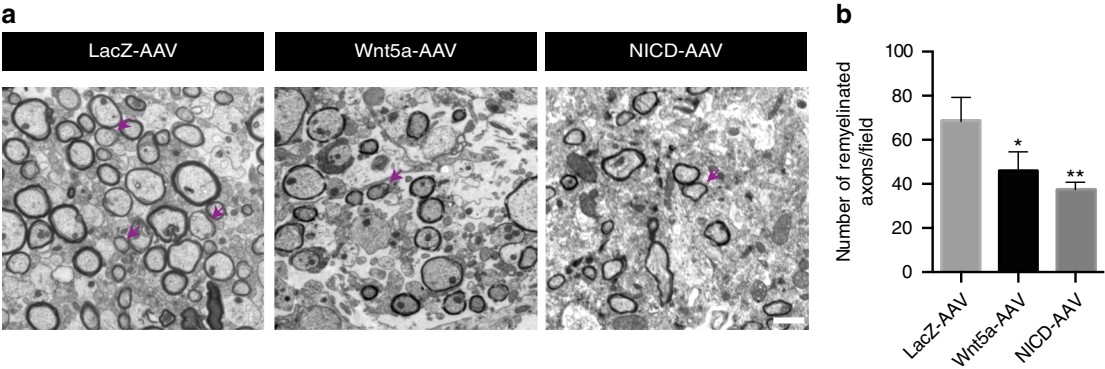

**Fig. 7** Forced activation of non-canonical Wnt or Notch signaling in the SVZ delays remyelination. **a** Electron microscopy images from demyelinated lesions of the SCWM reveal significantly fewer remyelinated axons in mice whose SVZ were infected with Wnt5a or NICD-AAV as compared to that of LacZ-AAV at 14DPI. Arrows indicate remyelinated axons. Scale bar represents 2 μm. **b** Quantification of remyelinated axons from LacZ-, Wnt5a-, and NICD-AAV-infected mice. Remyelinated axons were counted from fields of 300 μm². *$P < 0.05$, **$p < 0.01$; error bars represent mean ± s.e.m.; $n = 3$ independent experiments for **b**

annexin V staining, single cells were isolated as above and incubated in 100 μl of annexin V binding buffer (10 mM HEPES, 140 mM NaCl, and 2.5 mM CaCl₂, at pH 7.4) along with 5 μl of annexin V–Allophycocyanin (A35110, Thermo Fisher) and propidium iodide for 15 min at room temperature and then analyzed.

**Neurosphere and NSC cultures.** FACS SVZ *GFAP::GFP*+ cells were plated at clonal dilution (1 cell/μl) in floating non-adhesive 6-well culture dishes using neural stem cell medium (1:1 DMEM/F12 medium, 1× B27 supplement, 1× antibiotics, and daily addition of basic fibroblast growth factor at a concentration of 10 ng/ml)[69]. Primary neurospheres were mechanically dissociated into single cells after 7 days in vitro to form secondary neurospheres using the above culture conditions. Analysis was performed on secondary neurospheres. Single cells from secondary neurospheres were used to assay the effect of Wnt3a and Wnt5a on NSC polarization. Neurospheres were dissociated into single cells and plated (~500 cells/coverslip) on coverslips previously coated with poly-L-lysine (Sigma) and laminin (Sigma). Cells were cultured for 24 h in NSC culture medium and growth factors as above. For Wnt5a conditioned media stimulation, NSCs were cultured in the presence of Wnt5a and Wnt5a+CASIN 5 μM for an additional 16 h after which they were fixed in 4% paraformaldehyde (PFA) for 10 min at room temperature. For NSC transduction, LacZ, Wnt5a, or Wnt3a-AAV particles (0.25 μl, 1 × 10¹¹ viral particles/ml) were added to the cultures using fresh NSC medium. After 24 h, media with viral particles was replaced with fresh NSC medium, and cells were cultured for additional 3 days in vitro before analysis for β-catenin translocation by immunocytochemistry.

**Immunocytochemistry.** Neural stem cells on coverslips were fixed after the respective treatments in each experiment with 4% PFA at room temperature for 10 min, then washed with phosphate-buffered saline (PBS) twice. Coverslips were blocked with 10% goat serum (Invitrogen) containing 0.1% Triton X-100 and 1% bovine serum albumin (Sigma) in PBS for 10 min. Next, coverslips were incubated with primary antibodies at the indicated concentrations (see Supplementary Data 3 for a full list of antibodies, dilutions, and catalog numbers) at room temperature for 1 h and washed three times before incubation with the appropriate secondary antibodies (Jackson Immunoresearch; used at a 1:500 dilution for 1 h at room temperature). At the conclusion of this period coverslips were washed in PBS, stained with 4',6-diamidino-2-phenylindole to label nuclei, and mounted on microscope slides using MOWIOL.

**Long-term BrdU retention.** Long-term BrdU retention was used to detect LRCs (slow-cycling/quiescent NSCs) in the SVZ[43]. Briefly, mice were injected with BrdU (60 mg/kg) five times, 3 h apart, and then subject to a chase period of 30 days. Mice that received BrdU were then injected with Vehicle (DMSO; referred to as CTRL) or CASIN (Fig. 3l), or subjected to focal demyelination (Fig. 4h).

**Viral infection of the SVZ.** For adenoviral infection of the SVZ, 1 μl of AAV particles (1 × 10¹¹ viral particles/ml) driving expression of LacZ, Wnt3a, Wnt5a, or NICD were directly injected into the lateral ventricle (0, 1.8, 3.0 mm, anterioposterior, mediolateral, dorsoventral respective to Bregma). Mice were processed for analysis at 7 days post injury/AAV infection (See Fig. 6a, b and Supplementary Fig. 7a, b). LacZ-AAV was used as control, but it was also co-infected along with Wnt3a, Wnt5a, and NICD-AAVs (1:3 parts) to determine efficacy of SVZ/LV infection.

**Immunohistochemistry.** Mice were transcardially perfused with PBS (10 ml) followed by fixative solution (4% PFA in PBS; 20 ml) and brains were dissected and post-fixed for an additional 24 h. After 24 h of post-fixation, brains were cryo-protected in 30% sucrose for additional 24 h before sectioning. Free-floating brain sections (30 μm thick) containing the SVZ were blocked with 10% goat serum (Invitrogen) containing 0.3% Triton X-100 and 1% bovine serum albumin (Sigma) in PBS for 1 h at room temperature. Tissue sections were then incubated with primary antibodies at the indicated concentrations (see Supplementary Data 3 for a full list of antibodies, dilutions, and catalog numbers) overnight at 4 °C. When required, antigen retrieval was used for ki67 and PCNA immunostaining, for which tissue sections were incubated with 10 mM sodium citrate (pH 6.0) at 95 °C for 10 min and then rinsed twice with ice-cold PBS. For BrdU detection, tissue sections were treated with 2N HCL at 37 °C for 30 min, and then washed three times for 10 min each with ice-cold PBS before blocking. The appropriate secondary antibodies were purchased from Jackson Immunoresearch and used at a 1:500 dilution.

**Microscopy and cell counting.** A confocal laser-scanning microscope (TCS-SP5) (Leica DMI6000 B instrument) was used for image localization of FITC, CY3, and Cy5. Optical sections of confocal epifluorescence images were sequentially acquired using 20×, 40×, or 63× objectives (NA = 1.40), with LAS AF software. NIH ImageJ software was then used to merge images and cell counting was performed blindly from each experimental condition and, finally, compared with the respective controls. Tissue sections containing the SVZ were matched across samples in each experimental condition. For SVZ analysis, typically 4–8 correlative sections located between 1.045 and −0.08 mm from Bregma were analyzed. Additionally, the sections were manually assessed for the SCWM lesion in all experiments where demyelination was expected. All cell counts were performed along the length of SVZ in an area of 1 mm width that covers the ventricular surface. All cell quantification data were obtained by using the cell counter plugin on ImageJ.

**SVZ wholemount dissection and analysis.** SVZ wholemounts were dissected from adult mouse brains at 3DPI and 7DPI after lysolecithin injection[52]. Mice were deeply anesthetized with isoflurane, and the brain was dissected into ice-cold D1 media (1× HBSS, 1× antibiotics, 26 mM HEPES, 0.3% glucose, and 0.75% sucrose). The lateral ventricle was dissected after removing the hippocampus and septum. The wholemounts were fixed with 4% PFA in PBS overnight at 4 °C. On the second day, the underlying parenchyma was further dissected carefully. For the quantifications of apical *GFAP::GFP* cells, images were obtained from various anterior–ventral and posterior–dorsal regions, as these areas harbor clusters of NSCs[70].

**Electron microscopy analysis of demyelinated lesions.** Mice were perfused transcardially with PBS (pH 7.4) followed by 2% PFA and 2.5 % EM grade glutaraldehyde in 0.1 M phosphate buffer and post-fixed in the same solution overnight. The area of interest (corpus callosum) was dissected and placed in 2% osmium tetroxide in 0.1 M PBS (pH 7.4), dehydrated in a graded series of ethyl alcohol, and embedded with Durcupan resin in between two pieces of ACLAR sheets (EMS). Ultrathin sections of 80 nm were cut with a LEICA EM UC7 ultramicrotome and placed on formvar-coated slot copper grids. Sections were then counterstained with uranyl acetate and lead citrate and viewed with a FEI Tecnai12 BioTwinG2 electron microscope. Digital images were acquired with an AMT XR-60 CCD Digital Camera system.

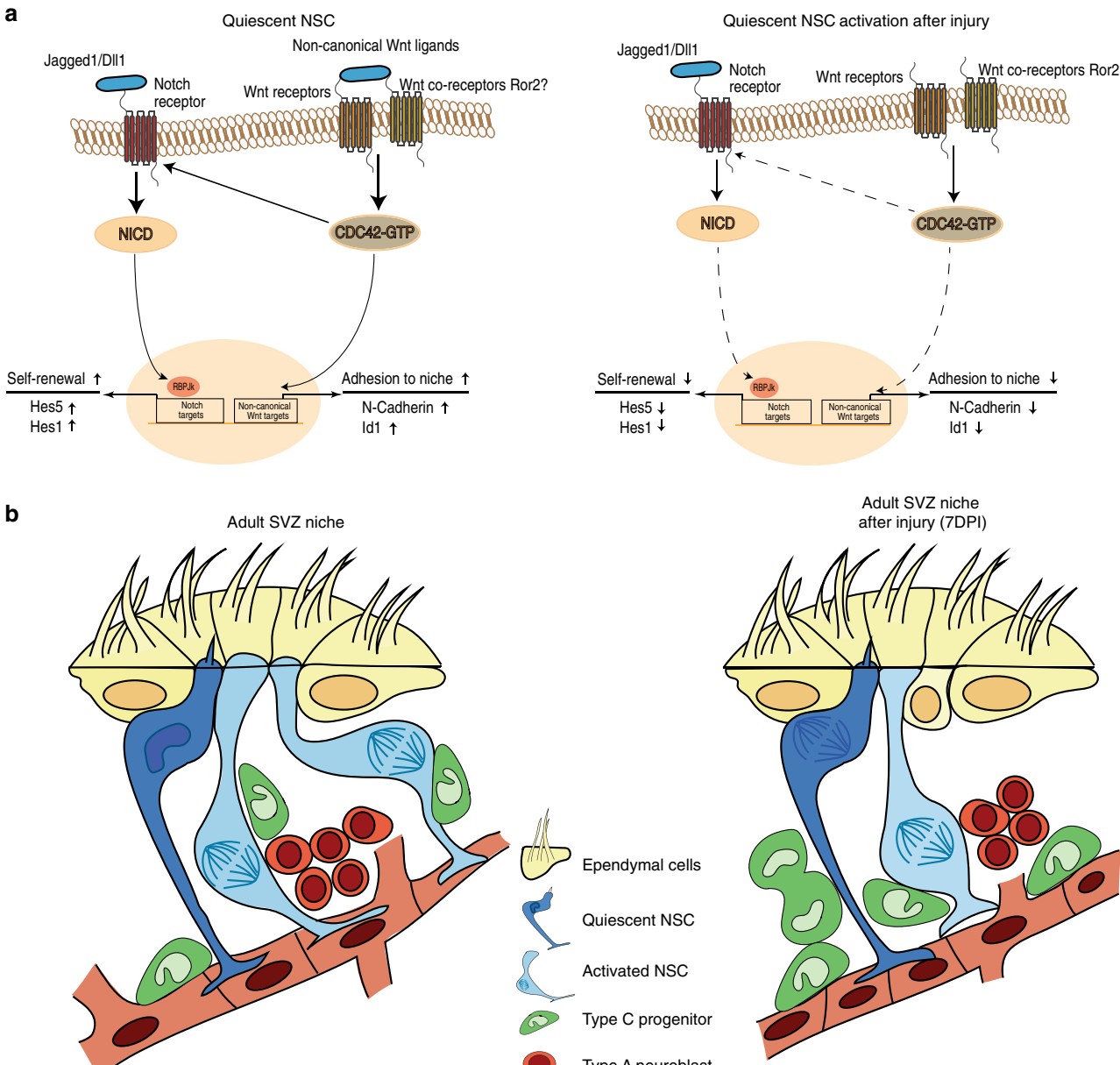

**Fig. 8** Proposed mechanism for maintenance of NSC quiescence in the SVZ during homeostasis and activation after injury. **a** Model of quiescent NSC maintenance under homeostatic conditions and after a demyelination injury. The quiescent state is maintained by non-canonical Wnt signaling-induced Cdc42 activity and its downstream targets Notch1, Id1, and N-Cadherin, thereby maintaining NSC self-renewal and anchorage to the SVZ niche. During a demyelination injury, a decrease in non-canonical Wnt ligand expression leads to a decrease in Cdc42 activity which disrupts Notch signaling and adhesion of qNSCs to the SVZ niche, which then undergo increased activation and lineage progression to generate progenitors. **b** Cytoarchitecture of the SVZ in normal physiology and after a demyelination injury (7DPI). During a demyelination injury, quiescent NSCs enter the cell cycle to undergo increased lineage progression and generate higher numbers of transit-amplifying progenitors that initiate the process of tissue repair

**Compound action potential recordings**. CAP recordings were registered as previously described[50] at 7DPI and 18DPI. Briefly, coronal brain slices (350 μm thick) were prepared in ice-cold artificial cerebrospinal fluid (ACSF) using the Leica VT1000S vibratome (Leica Biosystems, Germany). ACSF contained (in mM): 126 NaCl, 3 KCl, 1 $NaH_2PO_4$, 25 $NaHCO_3$, 1.3 $MgSO_4$, 2.4 $CaCl_2$, and 14 dextrose, and was saturated with 95%$O_2$/ 5%$CO_2$ gas mixture throughout the recordings. Injections of 1% of lysolecithin into the corpus callosum (CC; as above) lead to a consistent 400–500 μm focal demyelination. Before recording, brain slices containing focal demyelination in the CC were selected using a Kohler microscope with a lamp and a fiber optic light guide. One or two brain slices containing the CC focal lesion were used from each experimental animal. Slices were transferred to a recovery chamber filled with ASCF at room temperature (25 ± 0.5 °C). After ≥1 h of recovery time, a single slice was transferred to the recording chamber mounted under an upright microscope (Olympus BX51WI, Olympus, Japan) equipped with a CCD camera (CohuHD, US). The area of the lesion was identified under visual guidance with a 5× objective. Due to the absence of myelin, the site of lysolecithin injection had higher brightness in comparison with the surrounding healthy white matter tract[49]. Detailed investigations with a 40× water-immersion objective, differential interference contrast optic, and infrared illumination were used to verify the extent of the lesion.

**Statistics**. All statistical analyses were performed using Graphpad Prism 6 software. Data from all the experiments are expressed as mean ± s.e.m. All experimental groups were compared using the unpaired two-tailed $t$-test unless specified otherwise, and considered significant when $p < 0.05$.

**Data availability**. All the supporting data for this study are available from the corresponding author on reasonable request.

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

## Acknowledgements

We are very grateful to Dr. Ken-Ichi Takemaru (Pharmacology Department) and Dr. Joel Levine (Department of Neurobiology & Behavior) from SUNY, Stony Brook, for the Wnt3a and Wnt5a-AAV constructs and CTRL, Wnt3a, and Wnt5a cell lines. We are grateful to Susan Van Horn from the Central Microscopy Imaging Center at SUNY, Stony Brook, for her assistance in the electron microscopy studies. We would also like to thank Todd Rueb and Rebecca Connor of the Stony Brook University Flow Cytometry Facility for assistance with cell sorting. This work was supported by the following grants: NIMH RO1 RMH099384A (to A.A.; NIH) and RR-1601–07425 (to A.A; NMSS). M.K. acknowledges NIH Grant F31 NS093935.

## Author contributions

M.C. and A.A. conceptualized the experiments; M.C. and A.A. designed the methodology; M.C., M.K., and A.G.K. performed the experiments; M.C. and M.K. performed the electron microscopy experiments. S.F. assisted M.C. with some of the experiments. M.C., M.K., and A.A. analyzed the data; Y.G. and A.M. performed the compound action potential studies and analyzed the data; M.C. and A.A. wrote the manuscript.

## Additional information

**Competing interests:** The authors declare no competing financial interests.

