## [Peer Review File · Nature Communications]

Reviewers' comments:

Reviewer #1 (Remarks to the Author):

The authors investigate the signalling pathways and downstream effector that play a role in the control of NSCs activation vs quiescence. To this end, they use of model of demyelination/remyelination based on Lysolecithin injection in the corpus callosum of adult mice, a model which leads to the recruitment of both white matter and SVZ progenitors. Analysis of a previously produced proteomic datasets identified a number of proteins up- and down-regulated following demyelination. Building up onto this analysis, they demonstrate a role of the non-canonical Wnt-pathway, as well as of its downstream effector cdc42 in maintaining quiescence by controlling NSCs polarity, position and notch signalling. They finally show that post injury remyelination can be promoted by reducing notch activity.

This manuscript is novel and well written. It elegantly assembles different observations by integrating canonical and non-canonical Wnt signalling, and their putative role in controlling NSCs activation and lineage progression. The manuscript however suffers from several technical concerns or shortcuts in the reasoning which somehow weaken some of the conclusions.

Major concerns:

Label retaining protocol and quantifications: the length of BrdU administration is particularly short in the LRC protocol and only results in the labelling of a small population of cells. This can be appreciated on all photomicrographs (Fig 3 and 5, Suppl Fig 3). When facing such a low number of cells, state of the art stereological techniques should be used. Inaccurate quantification on small cell number otherwise results in variability which compromise conclusions. The variability of the quantifications can be best appreciated on Fig5f in this manuscript by comparing the control group. In this graph, while normalisation was achieved on the first control group, variability is very large on the two other control groups. In addition, co-expression of BrdU with other markers (GFAP, Sox2) is used to identify NSCs. Beside further reducing the number of cells to be quantified, it should be mentioned that analysis of GFAP is often complicated by 1) the cytoplasmic and diffuse expression of this marker, 2) variability in immunostaining quality, 3) changes in expression levels following treatment (one or the other explaining for example the dramatic GFAP loss observed in Fig 3i). As most of the conclusions of this manuscript are based on these quantifications, additional analysis using other marker combinations (or possibly excluding GFAP which I believe is not required) and/or appropriate stereological approach are needed.

The systematic normalization of quantification throughout the manuscript is another matter of concerns. Usually, this is done when methods that show large variability are used (e.g. tract tracers, viruses...) or when comparing experimental groups at different stages of a dynamic process. Methodological variability should not be a concern in the described experiments and it should be possible to present the raw numbers in the quantifications presented in this manuscript. This would provide important information on the consistency of the results and strength of the observations.

AAV: viruses are used in two instances: 1) to overexpress canonical vs. non canonical Wnts, 2) to overexpress NICD. To this end, AAVs are infused in the lateral ventricle. There is no direct analysis of the rate of transduction, dynamic of transgene expression (7 days between AAV injection and collection of the tissue, seem very short for AAVs) and identity of the transduced cells (ependymal or NSCs?). A better characterisation of the approach is needed to confirm efficient expression of the transgenes in the niche (Wnt transgenes) and most importantly in NSCs (NICD).

Assessment of functional repair following Notch activation: While most of the manuscript focuses on

non- canonical Wnt signalling and its target cdc42, the last part of this manuscript switches to Notch signalling to demonstrate relevance in myelin repair. Although this is acceptable because of the many "links" existing between the two pathways (and because of the qPCR data shown in Fig4c), this should be complemented by experiments focusing onto the non-canonical Wnt pathway and aimed at interfering with repair. While experiments aimed at activating the non-canonical Wnt pathway are difficult to perform, experiments aimed at preventing the disruption of non-canonical Wnt signalling in the SVZ during injury could be performed to confirm blockade of the endogenous remyelination process. This would further support the final conclusion, i.e. "These results directly demonstrate that the Notch signalling is sufficient to maintain NSC self-renewal and identity and that decrease in its activity in a non-canonical Wnt dependent manner is important in promoting the functional regeneration post-injury" which appears to currently be overstated.

Minor concerns:

Proteomic analysis: Proteins upregulated by 1.2 fold and downregulated by 0.8 fold were identified. How were these fold changes calculated? What is the confidence that they are significant/above noise level.

Blots quantifications:

- Figure 3 a, b: please indicate relative to what.
- Please show quantification for Fig 3g and Suppl Fig 2d blots

Methods description:

- It is often unclear where quantifications were performed (entire SVZ, dorsal most SVZ regions...). Please clarify.
- The methods for SVZ NSCs cultures and their in vitro stimulation with CASIN are not described
- The methods for luciferase assays are not described

Others:

- Line 163: only transcripts of the receptor *ror2* were upregulated in SVZ NSCs. It is unclear to which samples the authors refer, as only the SVZ was microdissected and NSCs were not purified.
- Line 197: the authors should refer to Fig 3c instead of Fig 2h?
- Ex vivo experiments should be repeated following demyelination to investigate change of dynamic interplay between canonical and non-canonical Wnt signalling.
- Activity of the canonical and non-canonical Wnt pathways were analysed following demyelination. How was the "active b-catenin assessed (nuclear? Phosphorylated?). A quantification should be provided.
- The increased VCAM1+ structures on the apical surface of SVZ whole mounts following CASIN treatment is difficult to interpret, as it suggests increase polarity and contact of NSCs (and therefore increase quiescence). Please comment.

It is unclear why authors have excluded the choroid plexus for analysing the changes in Wnt ligands expression following demyelination.

- Lines 343-350: it is unclear if b-catenin translocation was induced in vivo or in vitro.
- Cell sorting of EGFR sorted cells should be done in parallel to GFAP::GFP+ cells in order to confirm activation and lineage progression of the NSCs following demyelination.
- Line 387: please correct "transnationally antagonize »

Reviewer #2 (Remarks to the Author):

The present manuscript addresses the potential role of non-conventional wnt signaling in preserving the quiescent state of neural stem cells (NSCs) of the adult subventricular zone (SVZ). They show that

this pathway is associated to quiescence through activation of Cdc42 and subsequent activation of cell adhesion molecules, Id1 and Notch target genes. They also show that in response to a demyelinating injury (injection of lysolecithin into de corpus callosum) non-canonical signaling becomes reduced and NSCs become activated. This is an interesting paper but has a number of issues that the authors need to address, that are listed below.

In Figure 1 the authors contribute data relative to a proteomic analysis of SVZ tissue isolated from animals injected into the cerebral cortex with lysolecithin to induce demyelination in the corpus callosum or saline solution as a control. These data were generated in the context of a previous work of the same lab published in *The Journal of Neuroscience*. It is not clear why these data were not included in that previously published work. Now, the authors use this analysis as a starting point to choose non-canonical wnt signaling as a potentially relevant pathway for the response to the type of injury that they are studying. The problem is that, according to the provided information, wnt is not particularly highlighted above other signaling pathways in the proteomic analysis. Regarding Figure 1:

- The data presented are not informative about the rationale of the manuscript.
- Figures 1d & e especially do not add to the rationale of the work. Some proteins are listed but none has anything to do with the hypothesis that non-canonical wnt signaling would be a relevant hit to analyze.
- Moreover, Figures 1d & e show proteins identified in the proteomic analysis that were also found enriched in previous screenings using RNASeq in sorted qNSCs or aNSCs by other labs. First, the authors should explain the characteristics of these sorted cells (from the work by Codega et al.) as there have been other sorting experiments published. Second, none of the molecular candidates that are going to be analyzed in the manuscript appear in the list. Moreover, this piece of information would be better placed in Supplementary Information as it is a comparison with previously published data.

In Figure 2 the authors show a comprehensive analysis of different wnt molecules, as well as their inhibitors and co-receptors that are expressed in the SVZ relative to expression levels in the choroid plexus. Then, they perform an in vitro experiment in which they administer wnt3a (not expressed in the SVZ; canonical wnt signaling), wnt5a (non-canonical wnt signaling) or both produced by L-cell lines and use, as a readout, the protein levels of Axin2, a downstream target gene of canonical wnt signaling. This experiment shows that non-canonical wnt activation can antagonize the activity of the TCF/LEF-dependent wnt pathway. Regarding Figure 2:

- The expression of the different wnt and wnt-related molecules in the SVZ is shown as their value relative to that of the choroid plexus. With this type of analysis it is difficult to know which molecules are more or less abundantly expressed in the SVZs that are cultured in isolation.
- From a conceptual point of view, the choroid plexus is also an important, and strongly emerging, source of molecules with direct actions in SVZ NSCs. The authors explain in the text which are the wnts that more abundant in the choroid plexus but there is no quantitation of that. This is important information for the interpretation of in vivo experiments.

The authors then move to show how non-canonical wnt signaling induces the activation of Rho GTPase Cdc42, which is involved in apical cell polarity and adhesion, by stimulating SVZ explants with wnt5a with and without CASIN, a pharmacological inhibitor of Cdc42, and using the levels of GTP-bound Cdc42, the levels of Axin2 protein and Golgi polarization by immunocytochemistry with Golgi marker gm130 as readouts of activation/inhibition. The authors then show that treatment with wnt5a promotes the expression of Notch1, Id1 and N-cadherin in a Cdc42-dependent manner. The authors then inject the Cdc42 inhibitor in vivo to test the involvement of this GTPase in NSC activation and find fewer GFAP+SOX2+ BrdU-label retaining cells (LRCs). Regarding Figure 3:

- Panels of Figure 3c do not appear to be at the same magnification.
- Given that alpha-tubulin immunodetection looks different in the panels of Figure 3c, the authors should use an alternative protein as a loading control in their Western blots.
- The authors need to provide some quantitative data for observations shown in Figure 3c.
- The experiments with CASIN are highly variable. The authors need to increase sample size. Provided that wnt5a is expressed by SVZ cells (as shown in Figure 2), the authors need to rule out the effects of endogenous wnt5a. They need to include a condition with CASIN only. Part of the variability may be due to this endogenous source.
- Page 3 of Results section, line 173: the sentence "we also noticed that presence of both wnt3a and wnt5a suppressed axin2 expression further" should be rephrased.
- The experiment showing that CASIN also works in vivo is an important one but the authors only show one Western blot, with no quantification (Figure 3g). They need to provide more conclusive data that the inhibitor is acting in vivo. And the same for the data shown in Figure S2b. The number of B cells per pinwheel and the number of pinwheels are variable across the SVZ. The authors need to provide numbers and explain from where in the whole-mounts were the counts obtained.
- Most importantly, the experiment to evaluate the effect of the Cdc42 inhibitor in vivo is not optimal. It is extremely difficult to prove activation by showing dilution of the BrdU label in cells that divide slowly. This is suboptimal. The authors would need to show, for example, differences in Ki67 or MCM2 immunopositivity or perform dual nucleoside experiments (CldU/IdU). Or something similar to what the authors do in Figure 4. Also, considering the data obtained during injury (see Figure 4) the authors need to give an assessment about the position of the GFAP+SOX2+ LRCs (especially when considering the photographs shown in Figure 3i).
- The sentence in line 236 "The findings demonstrated that a continuous activation of Cdc2 is required to prevent premature activation of NSCs and their untimely exhaustion" is not supported by the data. It needs to be rephrased to a more accurate conclusion. Also, correct Cdc2.
- Very importantly, Cdc42 action in NSCs has already been characterized. The point of this work is to relate the regulation of this activity with a non-canonical wnt pathway in vivo. This important piece of the puzzle is not provided in the manuscript.

In Figure 4 the authors evaluate the effects of a lysolecithin treatment on wnt signaling and claim that there is a change from a non-canonical to a canonical activation of the wnt pathway, possibly determined by differential expression of wnt molecules that they show with PCR at different times during the injury protocol. The authors show that at the peak of demyelination Cdc42-GTP levels decrease and active beta-catenin and Axin 2 levels increase (Figure 4a). Then the authors perform a BrdU-labeling experiment to conclude that at the peak of demyelination NSCs are activated and dilute the label.

- The experiment shown in panel a is extremely important, yet no quantification is provided. Moreover, what is "active beta-catenin" mean?
- Define "diluted BrdU+ cells".
- In line 329, the sentence "From this analysis, we concluded that the decreased Cdc42 activity mediated by..." does not correctly reflect the results. The authors have not demonstrated that Cdc42 decreased activity and lowered non-canonical signaling mediate the effects of the lesion at this point in the manuscript.
- Again, in Figure 4h please explain where in the SVZ whole-mount were the counts taken.
- Have the authors tested specifically wnt7b and wnt11 in the assay shown in Figure 2e?

In Figure 5 the authors perform an important functional experiment in which they combine the lysolecithin-injury with the infection with AAVs carrying either wnt3a or wnt5a to show that during the lesion it is the downregulation of the non-canonical pathway which is responsible for the mobilization

of the NSCs. This is a key and difficult experiment. There are two issues for which some clarification would be desirable. First, the data on LRCs is shown as fold-change. It is well known that the numbers of LRCs in the SVZ are extremely low and therefore quantitations are difficult. Because of this, it is advisable to have another correlative set of data as, for example, the type of analysis shown in Figure S3c.

In Figure 6, the authors show that overexpression of Notch rescues the effects of lower activation of the non-canonical pathway. Why have the authors not used the type of analysis used in these experiments in the other experimental settings?

General points:

- It is disturbing that the authors use different types of analyses in the different experiments. For clarity and scholarly purposes they should homogenize the way they analyze the different experimental settings.
- The title should be changed to mention specifically that the injury that is analyzed is demyelination. Regeneration is now days a complex issue, the SVZ may respond differently to different types of injuries and therefore the title should be more specific.

Minor points:

- Legend of Figure 1: correct "atleast" (twice). In legend of Figure 3: correct "leftmost".
- What is SCWM in Methods ("Lysolecithin induced demyelination paradigm")?
- In the panels depicting in situ hybridization for different wnt elements shown in Figure S1a is very difficult to interpret what is a signal. Furthermore, it is not appropriate to include photographs taken from the Allen brain atlas as part of the results, even if they are supplementary data.
- In the legend of Figure S1, correct the sentence "error bars represent mean+SEM".
- Figure S2a is not quantitated and, although it is mentioned in the text, it is unclear what it represents.
- Line 197: is Fig. 3c not 2h.
- Improve labels in the flow cytometry plots.
- Define XAV939 somewhere (Figure S2).
- The authors could add the labels "canonical wnt ligands" and "non-canonical wnt ligands" to mRNA graphs in Figure 4.
- Line 387, please correct "post transnationally".
- Line 410, show the same number of decimals in the data.

Reviewer #3 (Remarks to the Author):

In this study, Chavali et al. report that the quiescent state of neural stem cells is maintained by a non-canonical Wnt signaling, through the activation of cdc42. Interestingly, after a demyelinating insult, this non-canonical Wnt signaling is decreased, and the authors infer that this is required to promote the differentiation of stem cells for repair. There are several interesting findings in this paper. Although Wnt pathways components have been described in stem cells, the distinction between canonical and non-canonical Wnt signaling is new, and adds a complexity to the regulation of quiescence in neural stem cells. This is a large and broad manuscript, and some of the conclusions are not clearly documented. The cellular localization of the many molecular changes is not always clear, for example. Also, at several points the authors do not clearly differentiate between quiescent stem cells and activated stem cells. In addition, the authors should make clear some of the quantitative analyses of cell numbers.

Specific Points:

1. "NSCs" were separated out as GFP+ in the GFAP:GFP mouse. This population should include both quiescent and activated stem cells. Activated stem cells are thought to differ from the quiescent by their expression of EGFR. Were these two populations separated to look at the expression of Wnt pathway genes or to look at the effects of Wnt signaling?
2. The SVZ also contains astrocytes. How were these factored into the various counting paradigms in the GFAP:GFP mice?
3. In Figure 1, were the NSCs the entire GFP+ population from the SVZ?
4. For Wnt signaling molecules the whole SVZ was studied, not separate populations. We don't know what cells in the SVZ express the various Wnt pathway components. For example, is Cdc42 expression confined to quiescent stem cells or is its expression also seen in the more mitotically active stem cells, or even in further derivatives?
5. The "polarization" promoted by Wnt 5a and shown in Figure 3c is not at all clear. In Figure 3c, why does the control Cdc42 merged with tubulin show cell processes, yet the GM130 merged with tubulin does not? It's difficult to see the difference between GM130 localization in controls and Wnt5a. Both have a paranuclear localization.
6. Wnt5a applied to SVZ explants induced expression of Notch1, Id1, and N-cadherin. The Id1 Western blot does not show much of a difference between Control and Wnt5a, however. A quantitation would be important here.
7. For the BrdU quantification in Figure 3, how many sections per mouse were examined to determine labeling index? Does this result suggest that the GFAP+/Sox3+ cells are the only ones that activate Cdc42 in response to Wnt5a?
8. The authors note that Cdc42 keeps cortical progenitors in cycle in an EGF dependent manner (line 238). I don't quite understand the point of this comparison, since quiescent NSCs don't express EGFR.
9. The authors postulate that blocking Cdc42 with CASIN results in activation of quiescent NSCs and therefore fewer BrdU+ quiescent NSCs. They find fewer BrdU+ cells. Another interpretation of this finding, however, is that blocking Cdc42 interferes with mitosis or cytokinesis (see Sakemori et al., JCI 2012 122:1052, who find that CASIN blocks cytokinesis of intestinal stem cells).
10. Do the authors have independent evidence that CASIN effectively blocks Cdc42 activity in this system?
11. In the experiments in which SVZs were infected with AAVs, why was the effect of Wnt3a the same as that of a LacZ virus?
12. Cdc42 induces the expression of a variety of adhesion molecules, although the authors provide no evidence that any of them functions to anchor NSCs to any particular cell type or matrix in the SVZ. The authors should not conclude definitively that these molecules are involved in anchorage. In a similar vein, the decrease of JAM-C and cadherins after a demyelinating insult is interesting, but it's not known in what cell(s) this decrease occurs.

Point-by-point response to reviewers' comments:

Reviewer #1 (Remarks to the Author):

The authors investigate the signaling pathways and downstream effector that play a role in the control of NSCs activation vs quiescence. To this end, they use of model of demyelination/remyelination based on Lysolecithin injection in the corpus callosum of adult mice, a model which leads to the recruitment of both white matter and SVZ progenitors. Analysis of a previously produced proteomic datasets identified a number of proteins up- and down-regulated following demyelination. Building up onto this analysis, they demonstrate a role of the non-canonical Wnt-pathway, as well as of its downstream effector cdc42 in maintaining quiescence by controlling NSCs polarity, position and notch signalling. They finally show that post injury remyelination can be promoted by reducing notch activity.

This manuscript is novel and well written. It elegantly assembles different observations by integrating canonical and non-canonical Wnt signalling, and their putative role in controlling NSCs activation and lineage progression. The manuscript however suffers from several technical concerns or shortcuts in the reasoning which somehow weaken some of the conclusions.

Major concerns:

1. Label retaining protocol and quantifications: the length of BrdU administration is particularly short in the LRC protocol and only results in the labelling of a small population of cells . This can be appreciated on all photomicrographs (Fig 3 and 5, Suppl Fig 3). When facing such a low number of cells, state of the art stereological techniques should be used. Inaccurate quantification on small cell number otherwise results in variability which compromise conclusions. The variability of the quantifications can be best appreciated on Fig5f in this manuscript by comparing the control group. In this graph, while normalisation was achieved on the first control group, variability is very large on the two other control groups. In addition, co-expression of BrdU with other markers (GFAP, Sox2) is used to identify NSCs. Beside further reducing the number of cells to be quantified, it should be mentioned that analysis of GFAP is often complicated by 1) the cytoplasmic and diffuse expression of this marker, 2) variability in immunostaining quality, 3) changes in expression levels following treatment (one or the other explaining for example the dramatic GFAP loss observed in Fig 3i). As most of the conclusions of this manuscript are based on these quantifications, additional analysis using other marker combinations (or possibly excluding GFAP which I believe is not required) and/or appropriate stereological approach are needed.

We appreciate the reviewer's encouraging and insightful comments. As suggested by the reviewer, we excluded the GFAP and Sox2 markers in our label retaining cell analyses. We are now presenting the absolute numbers of BrdU label retaining cells that were counted in the entire length of the SVZ in each section, in all of the experiments. We also performed additional experiments to increase the sample size to overcome the issue of variability that might arise from the fact that quiescent neural stem cells in the SVZ are fewer in numbers. We would also like to point out that this paradigm (5xBrdU injections, 3 hours apart) has been successfully utilized for reliably identifying the label retaining quiescent NSCs in the SVZ¹⁻³.

In addition, here we present FAC sorting data of GFAP::GFP cells in conjunction with CD15 (Lex) to identify quiescent NSC population^{4,5}. This data further supports our finding that quiescent NSCs (GFAP::GFP+CD15+ cells, see below) undergo transient activation and that their numbers are decreased as they give rise to increased numbers of transit amplifying progenitors (GFAP::GFP-CD15+ cells) in an effort initiate the lesion repair process after a demyelinating injury.

Reviewer Figure 1: FACS-purification of SVZ NSCs from GFAP::GFP mice 7 days after lysolecithin induced demyelination. (a-b) Typical FAC-sorting plot for CD15+GFP+ NSCs from GFAP::GFAP mice from the SVZ of NaCl injected (contralateral; a) and lysolecithin injected (Ipsilateral; b) hemispheres, 7 days after injection. Note that the total number of NSCs (CD15+GFP+ cells) are reduced, while NPC (CD15+GFP-) numbers are increased in the SVZ from the ipsilateral (lysolecithin) side. Lex (CD15) antigen was scattered with R-phycoerythrin, R-PE. Events scored = 30,000. A total number of 4 brains were pooled for this experiment.

2. The systematic normalization of quantification throughout the manuscript is another matter of concerns. Usually, this is done when methods that show large variability are used (e.g. tact tracers, viruses...) or when comparing experimental groups at different

stages of a dynamic process. Methodological variability should not be a concern in the described experiments and it should be possible to present the raw numbers in the quantifications presented in this manuscript. This would provide important information on the consistency of the results and strength of the observations.

In our study, we used several experimental approaches to analyze activation of SVZ quiescent NSCs and their participation in remyelination following a demyelination injury, including western blot, qPCR, confocal and electron microscopy, as well as compound action potential recordings, among others. Given the differing outputs (cell counts, c(t) values, band intensities, etc.) from these experiments, we normalized the data so that readers could easily cross-compare between different experiments (i.e., the control group is always set to “1”) without having to convert between different readouts. However, as per reviewer’s suggestion, we agree that for the cell counts it is more useful to present the raw data. Therefore, we are now presenting the data of absolute numbers of cells quantified in all the pertaining data sets and the average Ct values for the qPCR data presented in Figure 2.

3. AAV: viruses are used in two instances: 1) to overexpress canonical vs. non canonical Wnts, 2) to overexpress NICD. To this end, AAVs are infused in the lateral ventricle. There is no direct analysis of the rate of transduction, dynamic of transgene expression (7 days between AAV injection and collection of the tissue, seem very short for AAVs) and identity of the transduced cells (ependymal or NSCs?). A better characterisation of the approach is needed to confirm efficient expression of the transgenes in the niche (Wnt transgenes) and most importantly in NSCs (NICD).

We thank the reviewer for this suggestion. We are now providing additional analysis to validate the adenoviral transduction of Wnt and NICD constructs in the SVZ. In the revised manuscript, we provide high magnification images that demonstrate the infection of the SVZ NSCs (Supplementary Figure 6 and Supplementary Figure 7). It must be noted that since there are no reliable commercially available antibodies against Wnt5a and Wnt3a, the immunohistochemical analysis of these Wnt ligands could not be performed to verify their expression after AAV infection. We therefore used the Wnt signaling downstream target *Axin2* to verify the activation of the non-canonical and canonical Wnt pathways after infection with Wnt5a and Wnt3a-AAV respectively (*Axin2* is repressed by Wnt5a, and activated by Wnt3a – see Figure 2g-h). To this end, we now provide evidence for efficient targeting of Wnt AAVs by evaluating *Axin2*⁺ expression in GFAP⁺ NSCs in the SVZ. An evident decrease in GFAP⁺*Axin2*⁺ cells in the SVZ was observed 7 days after the Wnt5a-AAV infection and an increase in the same population was observed after infection with Wnt3a-AAV, as compared to the LacZ-AAV condition (Supplementary Figure 6c-e). We also confirmed this further by analyzing *Axin2* mRNA levels in the SVZ 7 days after the LacZ, Wnt5a, and Wnt3a infection (Supplementary Figure 6f-g). In addition, we also provide evidence for targeting of NICD-AAV in the SVZ NSCs by immunohistochemistry (Supplementary Figure 7c-d) and further validated this data by analyzing levels of the NICD protein, and mRNA levels of the Notch downstream target *Hes5* in the SVZ 7 days after NICD-AAV infection (Supplementary Figure 7e-f).

4. Assessment of functional repair following Notch activation: While most of the manuscript focuses on non- canonical Wnt signalling and its target *cdc42*, the last part of this manuscript switches to Notch signalling to demonstrate relevance in myelin repair. Although this is acceptable because of the many “links” existing between the two pathways (and because of the qPCR data shown in Fig4c), this should be complemented by experiments focusing onto the non-canonical Wnt pathway and aimed at interfering

with repair. While experiments aimed at activating the non-canonical Wnt pathway are difficult to perform, experiments aimed at preventing the disruption of non-canonical Wnt signalling in the SVZ during injury could be performed to confirm blockade of the endogenous remyelination process. This would further support the conclusion, i.e. “These results directly demonstrate that the Notch signalling is sufficient to maintain NSC self-renewal and identity and that decrease in its activity in a non-canonical Wnt dependent manner is important in promoting the functional regeneration post-injury” which appears to currently be overstated.

We thank the reviewer for this thoughtful suggestion. To address this point, we performed a non-canonical Wnt gain-of-function experiment in the SVZ using the Wnt5a-AAV during a demyelination injury (Figure 6a), and analyzed the remyelination outcome by electron microscopy imaging at 14 days post injury. This new data set (Figure 7a-b) shows that activation of non-canonical Wnt signaling in the SVZ is sufficient to delay remyelination after a demyelinating injury. We believe this data will further strengthen our claim that non-canonical Wnt activity can directly regulate NSC activation and lineage progression in the context of white matter injury and repair.

Minor concerns:

Proteomic analysis: Proteins upregulated by 1.2 fold and downregulated by 0.8 fold were identified. How were these fold changes calculated? What is the confidence that they are significant/above noise level.

The tandem mass spectra data was first analyzed using a binary classifier that is trained on a manually validated data set to remove the low-quality MS/MS spectra. The resulting spectra were analyzed against a mouse protein FASTA sequence database EBI-IPI⁶, and a decoy database with reversed sequences⁷ was used to calculate the confidence levels and the false positive rates. The spectra were further validated by DTAselect⁸. Furthermore, a minimum of 7 amino acid residues sequence length was required, and two independent peptide identifications was required to reduce the false discovery rate to less than 1%. Finally, Spectral counting methods^{9,10} were used to obtain the fold changes. This information is now added to the proteomics description in the methods section.

Blots quantifications:

- Figure 3 a, b: please indicate relative to what.

We have now added this data. The Cdc42-GTP levels in all the experiments were normalized to the total Cdc42 protein levels and this is indicated on the axis of all the plots showing these quantifications.

- Please show quantification for Fig 3g and Suppl Fig 2d blots.

We have now added quantifications of these blots.

Methods description:

- It is often unclear where quantifications were performed (entire SVZ, dorsal most SVZ regions...). Please clarify.

The entire SVZ has been analyzed for the quantifications we present. Specifically, for the analysis of frozen sections, images were obtained for quantification from the entire length of the lateral ventricular wall from at least 4-6 sections located between +1.045 and -0.08 mm from bregma were analyzed. In the SVZ wholemounts, GFAP::GFP cell images and numbers were obtained from at least 10 fields from, anterior-ventral and posterior-dorsal regions that have high clusters of NSCs¹¹. We have now added this information to the methods.

- The methods for SVZ NSCs cultures and their in vitro stimulation with CASIN are not described.

We apologize for this missing information and have now updated these methods and protocols.

- The methods for luciferase assays are not described

The methods section has been updated with the luciferase assay protocol and experimental details.

Others:

- Line 163: only transcripts of the receptor *ror2* were upregulated in SVZ NSCs. It is unclear to which samples the authors refer, as only the SVZ was microdissected and NSCs were not purified.

Figure 2d shows the qPCR analysis of Wnt co-receptors expression pattern in freshly sorted GFAP::GFP+ cells. This data shows that expression of Ror2 was significantly higher in the GFAP::GFP+ NSC population as compared to other SVZ-resident cells (the GFP negative population).

- Line 197: the authors should refer to Fig 3c instead of Fig 2h?

The figure numbers have now been corrected.

- Ex vivo experiments should be repeated following demyelination to investigate change of dynamic interplay between canonical and non-canonical Wnt signalling.

While this is an interesting suggestion, we would like to highlight the importance of niche derived signals that regulate the NSC properties *in vivo* after an injury. We believe the injury induced signals directly affect the SVZ cell populations only in their native environment. Given the complexity of the SVZ niche and the different components from which these signals emerge (for example: choroid plexus, vascular niche, etc.), mimicking this in our *ex vivo* system may not be possible and could also affect the molecular and cellular outcomes. Indeed, this is the reason why we chose to perform a rescue experiment *in vivo* using the lysolecithin model of demyelination to confirm the significance of non-canonical Wnt signaling in NSC maintenance (Figure 6).

- Activity of the canonical and non-canonical Wnt pathways were analysed following demyelination. How was the “active β -catenin assessed (nuclear? Phosphorylated?). A quantification should be provided.

We added quantifications for these blots (Figure 4b-c, e-f). In our manuscript, we refer to Active β -Catenin as the stabilized form of β -catenin that has not been phosphorylated by GSK-3 β , and is functionally active in the canonical Wnt signaling pathway (Reference 6, also see Cell

Signaling Technology product 8814; Non-phospho (Active) β -Catenin (Ser33/37/Thr41)).

- The increased VCAM1+ structures on the apical surface of SVZ whole mounts following CASIN treatment is difficult to interpret, as it suggests increase polarity and contact of NSCs (and therefore increase quiescence). Please comment.

We thank the reviewer for this observation. We agree with the reviewer that this data is difficult to interpret and therefore we removed it from the revised manuscript.

-It is unclear why authors have excluded the choroid plexus for analysing the changes in Wnt ligands expression following demyelination.

Although interesting, this is an experimental limitation. Choroid plexus tissue from only one hemisphere (contralateral vs ipsilateral sides) constitutes very little tissue for the RNA analysis to achieve reproducible results. We have now added new text to reflect this in our discussion.

- Lines 343-350: it is unclear if b-catenin translocation was induced in vivo or in vitro.

We apologize for this issue. The nuclear translocation of β -catenin was assessed after LacZ, Wnt3a, or Wnt5a-AAV *in vitro* infection of primary NSCs obtained from the SVZ. We have now added this information to the methods section in the revised manuscript.

- Cell sorting of EGFR sorted cells should be done in parallel to GFAP::GFP+ cells in order to confirm activation and lineage progression of the NSCs following demyelination.

Although we did not perform FAC sorting of this population due to the variation in the EGFR levels in NSCs post demyelination injury, we are now providing the immunohistochemistry data that shows an increase in the GFAP-EGFR+ cell population in the SVZ, confirming the activation and lineage progression of NSCs in the SVZ during demyelination (Figure 5f-g). In addition, in reviewer figure 1, increased numbers of GFAP-CD15+ NPCs can be observed, further confirming an increase in NSC lineage progression in the SVZ after a demyelination injury.

- Line 387: please correct “transnationally antagonize

We apologize for this mistake, we fixed this issue and have thoroughly revised our text.

Reviewer #2 (Remarks to the Author):

The present manuscript addresses the potential role of non-conventional wnt signaling in preserving the quiescent state of neural stem cells (NSCs) of the adult subventricular zone (SVZ). They show that this pathway is associated to quiescence through activation of Cdc42 and subsequent activation of cell adhesion molecules, Id1 and Notch target genes. They also show that in response to a demyelinating injury (injection of lysolecithin into de corpus callosum) non-canonical signaling becomes reduced and NSCs become activated. This is an interesting paper but has a number of issues that the authors need to address, that are listed below.

In Figure 1 the authors contribute data relative to a proteomic analysis of SVZ tissue

isolated from animals injected into the cerebral cortex with lysolecithin to induce demyelination in the corpus callosum or saline solution as a control. These data were generated in the context of a previous work of the same lab published in *The Journal of Neuroscience*. It is not clear why these data were not included in that previously published work. Now, the authors use this analysis as a starting point to choose non-canonical wnt signaling as a potentially relevant pathway for the response to the type of injury that they are studying. The problem is that, according to the provided information, wnt is not particularly highlighted above other signaling pathways in the proteomic analysis.

- Regarding Figure 1: The data presented are not informative about the rationale of the manuscript.

We would like to thank the reviewer for his/her interest in our work and we appreciate the remarks. As described in the manuscript our proteomics screen revealed 790 proteins that were upregulated and 166 proteins that were downregulated in the SVZ following a demyelination injury. These proteins broadly belong to the pathways that are involved in metabolic changes, cell adhesion, and cell cycle regulation. Alterations in these pathways normally occur during periods of increased activation of normally dormant stem cells¹². From this data set we put forth the idea that an extensive activation occurs in the SVZ niche after a demyelination injury.

Considering the reviewer's comments, we reanalyzed the proteomics data using the REACTOME pathway database¹³. In addition to this new pathway analysis (Figure 1c), we also performed a STRING protein-protein interaction network analysis¹⁴ employing a high confidence evidence based approach with a minimum interaction score of 0.9, using the hits from our proteomics screen that were identified as a part of Wnt and Rho-GTPase signaling pathways. This analysis showed a high enrichment in interactions between these two pathways. The majority of the proteins in this network interacted highly with Cdc42 and β -catenin which are known to be a part of Wnt signaling and cell adhesion mechanisms (Figure 1d). These results point to alterations in both these pathways in the SVZ during a demyelination injury, which we focused on further using a combination of *in vitro* and *in vivo* models.

We believe this revised analysis demonstrates Wnt signaling as a relevant hit within this dataset, strengthening our rationale for further investigating the role of the Wnt pathway in maintenance of SVZ NSCs during a demyelination injury and the subsequent repair process.

- Figures 1d & e especially do not add to the rationale of the work. Some proteins are listed but none has anything to do with the hypothesis that non-canonical wnt signaling would be a relevant hit to analyze.

We agree with the reviewer on this point and revised this data to highlight the rationale of our study. Based on the STRING protein interaction analysis we provide relevant protein hits that are a part of Rho GTPase and Wnt pathways (Figure 1e). Wnt and Rho GTPase signaling activity are directly involved in the regulation of Cdc42 and β -catenin, which are important effectors of non-canonical and canonical Wnt signaling respectively¹⁵⁻²². We further analyzed the functional relevance of these two proteins in regulating the Wnt pathway in the SVZ NSCs in both homeostasis and after a demyelination injury.

- Moreover, Figures 1d & e show proteins identified in the proteomic analysis that were also found enriched in previous screenings using RNASeq in sorted qNSCs or aNSCs by other labs. First, the authors should explain the characteristics of these sorted cells

(from the work by Codega et al.) as there have been other sorting experiments published. Second, none of the molecular candidates that are going to be analyzed in the manuscript appear in the list. Moreover, this piece of information would be better placed in Supplementary Information as it is a comparison with previously published data.

We agree with the reviewer and have now revised this data that is presented in Figure 1 and supplementary figure 1. Furthermore, we added new text in the results section about the quiescent NSC population described by Codega et al.,²³. While we agree that proteins that were identified in our proteomics screen and enriched in qNSCs (Codega et al.) are not analyzed in our work, we would like to convey the message that alterations in the expression levels of these proteins during a demyelination injury suggests that they impact the properties of qNSCs. In addition, we have now added proteins identified in our screen that are directly linked to Cdc42 activity and β -catenin regulation (Figure 1d-e).

In Figure 2 the authors show a comprehensive analysis of different wnt molecules, as well as their inhibitors and co-receptors that are expressed in the SVZ relative to expression levels in the choroid plexus. Then, they perform an in vitro experiment in which they administer wnt3a (not expressed in the SVZ; canonical wnt signaling), wnt5a (non-canonical wnt signaling) or both produced by L-cell lines and use, as a readout, the protein levels of Axin2, a downstream target gene of canonical wnt signaling. This experiment shows that non-canonical wnt activation can antagonize the activity of the TCF/LEF-dependent wnt pathway. Regarding Figure 2:

- The expression of the different wnt and wnt-related molecules in the SVZ is shown as their value relative to that of the choroid plexus. With this type of analysis it is difficult to know which molecules are more or less abundantly expressed in the SVZs that are cultured in isolation.

We agree with the reviewer about the representation of our qPCR analysis. We are now presenting the average Ct values (normalized to *GAPDH*) for both choroid plexus and the SVZ tissue as a supplemental table (Supplementary table 2). The expression of SVZ Wnt ligands and Wnt inhibitors is represented relative to choroid plexus to determine the possible gradients in these factors. Wnt signals are short range and often activate their targets in cells that are in contact with each other²⁴. It is therefore important to identify the origin and the gradients of these signals in the SVZ microenvironment. Indeed, our analysis suggests that both non-canonical Wnt ligands and the canonical Wnt inhibitors are highly expressed in the choroid plexus tissue, which is in line with results from our work and previously published reports that canonical Wnt signaling is absent in the SVZ NSCs under homeostatic conditions^{1, 25}.

- From a conceptual point of view, the choroid plexus is also an important, and strongly emerging, source of molecules with direct actions in SVZ NSCs. The authors explain in the text which are the wnts that more abundant in the choroid plexus but there is no quantitation of that. This is important information for the interpretation of in vivo experiments.

We thank the reviewer for the opportunity to clarify this point. In the qPCR data presented in Figure 2a-c, we quantified the gene expression of Wnt ligands and inhibitors in both choroid plexus and SVZ tissue. As mentioned in the above comment, we are now providing the average Ct values as a supplemental table (Supplementary table 2), which will further clarify this issue.

The authors then move to show how non-canonical wnt signaling induces the activation

of Rho GTPase Cdc42, which is involved in apical cell polarity and adhesion, by stimulating SVZ explants with wnt5a with and without CASIN, a pharmacological inhibitor of Cdc42, and using the levels of GTP-bound Cdc42, the levels of Axin2 protein and Golgi polarization by immunocytochemistry with Golgi marker gm130 as readouts of activation/inhibition. The authors then show that treatment with wnt5a promotes the expression of Notch1, Id1 and N-cadherin in a Cdc42-dependent manner. The authors then inject the Cdc42 inhibitor in vivo to test the involvement of this GTPase in NSC activation and find fewer GFAP+SOX2+ BrdU-label retaining cells (LRCs). Regarding Figure 3:

- Panels of Figure 3c do not appear to be at the same magnification.

We have resized and reconfigured this figure with additional intensity distribution plots for both tubulin and Cdc42. We also provide new representative immunocytochemistry images labeled with antibodies against N-Cadherin (Figure 3d) to demonstrate that Wnt5a promotes polarization of NSCs via Cdc42 activation.

- Given that alpha-tubulin immunodetection looks different in the panels of Figure 3c, the authors should use an alternative protein as a loading control in their Western blots.

We used the protein lysates from these experiments and performed western blot analysis for beta actin (alternative loading control) and alpha tubulin. In Reviewer Figure 2, we quantified the levels of alpha tubulin and actin from four different protein lysates from these experiments, and it can be noted that there are no significant differences in alpha tubulin levels as compared to actin levels. Therefore, we kept alpha tubulin as our loading control.

Reviewer Figure 2: Immunoblot for alpha tubulin and actin from four different protein lysates. Graph showing the ratio of actin to tubulin levels. n=4 independent experiments.

- The authors need to provide some quantitative data for observations shown in Figure 3c.

We have now added an intensity distribution plot to quantitatively show the polarization of Cdc42 and alpha tubulin (Figure 3c).

- The experiments with CASIN are highly variable. The authors need to increase sample size. Provided that wnt5a is expressed by SVZ cells (as shown in Figure 2), the authors need to rule out the effects of endogenous wnt5a. They need to include a condition with CASIN only. Part of the variability may be due to this endogenous source.

We thank the reviewer for raising this important point. We have now performed additional experiments to increase the sample size to overcome the variation issue that might be due to endogenous Wnt5a. In this context, we also provide a comparison of Cdc42-GTP levels between Control and CASIN alone treated SVZ explants. As it can be observed, we did not detect any significant changes in the Cdc42-GTP levels between the Control and CASIN only treatments (Reviewer Figure 3).

Reviewer Figure 3: Cdc42-GTP levels after stimulation with CASIN only. SVZ explants treated with 5µM CASIN did not show any significant changes in the Cdc42-GTP levels as determined by Cdc42-GTP immunoblots. n=3 independent experiments

- Page 3 of Results section, line 173: the sentence “we also noticed that presence of both wnt3a and wnt5a suppressed axin2 expression further” should be rephrased.

In the revised manuscript, we rephrased this sentence, which now reads as“ Interestingly, in the presence of both ligands, Wnt5a antagonized the Wnt3a induced axin2 protein levels”.

- The experiment showing that CASIN also works in vivo is an important one but the authors only show one Western blot, with no quantification (Figure 3g). They need to provide more conclusive data that the inhibitor is acting in vivo. And the same for the data shown in Figure S2b. The number of B cells per pinwheel and the number of pinwheels are variable across the SVZ. The authors need to provide numbers and explain from where in the whole-mounts were the counts obtained.

We have now increased the sample size and quantified all the western blots (Cdc42-GTP, Notch1, Id1, and N-Cadherin) to show that CASIN acts *in vivo* as it does in our explant system. This data is now presented in Figure 3h-k. In addition to this, we have now also analyzed ki67+ proliferating cells in the SVZ (supplementary Figure 2c-d) to further confirm the effect of CASIN treatment on NSC activation. Finally, we removed the data set on the VCAM1+ pinwheel structures, as reviewer # 1 pointed out that the increase in VCAM1+ apical structures is difficult to interpret.

- Most importantly, the experiment to evaluate the effect of the Cdc42 inhibitor in vivo is not optimal. It is extremely difficult to prove activation by showing dilution of the BrdU label in cells that divide slowly. This is suboptimal. The authors would need to show, for example, differences in Ki67 or MCM2 immunopositivity or perform dual nucleoside experiments (CldU/IdU). Or something similar to what the authors do in Figure 4. Also, considering the data obtained during injury (see Figure 4) the authors need to give an assessment about the position of the GFAP+SOX2+ LRCs (especially when considering the photographs shown in Figure 3i).

This is an excellent suggestion. We have now performed ki67 immunostaining to determine whether activation and loss of the BrdU label is accompanied by an increase in proliferating cells in the SVZ. This data is now presented in supplementary figure 2c-d and shows an increase in the ki67+ cell numbers in the SVZ after CASIN treatment. This data in conjunction with the decrease in BrdU label retaining cell numbers and the decrease in Notch1, Id1, and N-Cadherin protein levels further confirms an increased activation of quiescent NSCs in the SVZ after inhibition of Cdc42 activity.

The reviewer also brings up an interesting point about the positioning of the BrdU label retaining cells after the CASIN treatment. We have quantified the distance of these cells from the ventricular wall and the data is presented here in reviewer figure 4.

Reviewer Figure 4: Distance of BrdU label retaining cells from the ventricular wall after CASIN treatment. n=25 cells from 4 animals for each condition.

As it can be seen from the above data, CASIN treatment induced a positional loss of BrdU label retaining cells as we observed an increase in the average distance of BrdU label retaining cells from the ventricular wall, albeit not significantly (7.85 ± 0.83 in vehicle vs 9.83 ± 1.37 µm after CASIN treatment). We believe this could be due to the transient inhibitory activity of CASIN. It might be possible that a stronger dosage and/or prolonged treatment might result in a significant displacement of qNSCs from their native niche.

- The sentence in line 236 “The findings demonstrated that a continuous activation of Cdc2 is required to prevent premature activation of NSCs and their untimely exhaustion” is not supported by the data. It needs to be rephrased to a more accurate conclusion. Also, correct Cdc2.

We agree with this comment, hence we rephrased this sentence which now as reads as.....
“These findings demonstrate that inhibiting Cdc42 activity disrupted the quiescent state of NSCs and triggered their activation”.

- Very importantly, Cdc42 action in NSCs has already been characterized. The point of this work is to relate the regulation of this activity with a non-canonical Wnt pathway in vivo. This important piece of the puzzle is not provided in the manuscript.

We thank the reviewer for this observation. We have now performed a Wnt5a-AAV gain-of-function rescue experiment along with the CASIN treatment to block Cdc42 activation in a

demyelination injury setting. In line with our original findings, we observed that CASIN treatment in Wnt5a infected mice does not rescue the loss of BrdU label retention capacity, showing Cdc42 acts downstream of Wnt5a (or non-canonical Wnt signaling) (Figure 6g-i). Indeed, we also detected that Wnt5a-AAV infection decreased ki67+ cell numbers in the SVZ during following a demyelination injury (Figure 6e-f). This cytostatic effect was also hindered after CASIN treatment, as Wnt5a-AAV infection along with CASIN treatment could not induce anti-proliferatory effect in the SVZ (Figure 6g-i). This data further supports our hypothesis that Wnt5a induced activation of Cdc42 is required for the maintenance of NSC quiescence.

In Figure 4 the authors evaluate the effects of a lysolecithin treatment on wnt signaling and claim that there is a change from a non-canonical to a canonical activation of the wnt pathway, possibly determined by differential expression of wnt molecules that they show with PCR at different times during the injury protocol. The authors show that at the peak of demyelination Cdc42-GTP levels decrease and active beta-catenin and Axin2 levels increase (Figure 4a). Then the authors perform a BrdU-labeling experiment to conclude that at the peak of demyelination NSCs are activated and dilute the label.

- The experiment shown in panel a is extremely important, yet no quantification is provided. Moreover, what is “active beta-catenin” mean?

We have now added quantifications that show a significant decrease in Cdc42-GTP levels, and an increase in active β -catenin levels (Figure 4b-c, e-f). We refer to active β -Catenin as the stabilized form of β -catenin that has not been phosphorylated by GSK-3 β , and is functionally active in the canonical Wnt signaling pathway²⁶. Its upregulation suggests an increase in β -catenin dependent canonical Wnt signaling.

- Define “diluted BrdU+ cells”.

As shown in our confocal images, BrdU intensity in quiescent NSCs of the SVZ is relatively strong at the end of the chase period (30 days) under normal conditions (Supplementary Figure 4a). However, this BrdU intensity in quiescent NSCs is reduced at the peak of demyelination (7 days post injury) compared to control quiescent NSCs, as they enter cell cycle and start to divide (diluting the BrdU label as a result; Supplementary Figure 4b). We have now added an intensity quantification to describe these diluted BrdU+ cells (Supplementary Figure 4c).

- In line 329, the sentence “From this analysis, we concluded that the decreased Cdc42 activity mediated by...” does not correctly reflect the results. The authors have not demonstrated that Cdc42 decreased activity and lowered non-canonical signaling mediate the effects of the lesion at this point in the manuscript.

This sentence is now rephrased as “This data suggests that a decrease in Cdc42 activity during an acute demyelination injury results in the downregulation of adhesion molecules that maintain NSC anchorage to the apical SVZ niche”.

- Again, in Figure 4h please explain where in the SVZ whole-mount were the counts taken.

For the quantifications of apical GFAP::GFP+ cell numbers, images were obtained from the SVZ wholemounts from anterior-ventral and posterior-dorsal regions that have been shown to harbor clusters of NSCs¹¹. We have now added this information to the methods section.

- Have the authors tested specifically wnt7b and wnt11 in the assay shown in Figure 2e?

While we did not test Wnt7b and Wnt11 in our assays, it is widely accepted that Wnt7b induces canonical β -catenin signaling^{22, 27} and Wnt11 induces non-canonical signaling by regulating Cdc42 activity^{15, 28}. The commercially available Wnt3a and Wnt5a L-cell lines are well characterized, and are commonly used to induce canonical and non-canonical Wnt signaling respectively. As such, we used those in our studies²⁹.

In Figure 5 the authors perform an important functional experiment in which they combine the lysolecithin-injury with the infection with AAVs carrying either wnt3a or wnt5a to show that during the lesion it is the downregulation of the non-canonical pathway which is responsible for the mobilization of the NSCs. This is a key and difficult experiment. There are two issues for which some clarification would be desirable. First, the data on LRCs is shown as fold-change. It is well known that the numbers of LRCs in the SVZ are extremely low and therefore quantitations are difficult. Because of this, it is advisable to have another correlative set of data as, for example, the type of analysis shown in Figure S3c.

We thank the reviewer for this suggestion. To address these issues, we are now providing the absolute numbers of BrdU label retaining cells quantified in the entire length of the SVZ from each coronal section in all experimental conditions. We also increased the sample size to address any variability that might arise from the lower numbers of LRCs in the SVZ (Figure 6c-d). Furthermore, as suggested by the reviewer, we have now performed ki67 immunostaining experiments to validate our results. In addition to rescuing the decrease in BrdU label retaining cells, we also observed a decrease in ki67+ cells in Wnt5a-AAV infected SVZ despite an acute demyelination injury (Figure 6e-f). This data further suggests that Wnt5a induced non-canonical Wnt signaling is crucial for the maintenance of the quiescent status in NSCs. These data sets are now a part of Figure 6.

In Figure 6, the authors show that overexpression of Notch rescues the effects of lower activation of the non-canonical pathway. Why have the authors not used the type of analysis used in these experiments in the other experimental settings?

We thank the reviewer for this observation. We reorganized the data that was presented in the Figure 6 of earlier version. We now performed additional experiments to homogenize the way we analyzed these experimental settings (Wnt5a and NICD-AAV infection experiments). Using Wnt5a-AAV infection we force induced non-canonical Wnt signaling in the SVZ during a demyelination injury and assessed the remyelination at 14 days' post injury by electron microscopy. This data is now presented in Figure 7a-b.

General points:

- It is disturbing that the authors use different types of analyses in the different experiments. For clarity and scholarly purposes they should homogenize the way they analyze the different experimental settings.

We thank the reviewer for this suggestion. In addition to the above point, we have reorganized our data and performed additional experiments to homogenize our data and analysis to focus on the role of Wnt and Cdc42 signaling and its downstream targets in the maintenance of SVZ NSCs in homeostatic conditions and after a demyelination injury, which we consider to be the

primary focus of our study. As Notch is downstream target of non-canonical Wnt signaling in the SVZ NSCs, we show an additional set of functional data in the context of Wnt/Cdc42/Notch signaling axis (Figure 7c-d). We have now added new text to reflect this in the results section describing these experiments. We believe the revised manuscript is now homogenous in the way the results were analyzed for various experimental settings.

- The title should be changed to mention specifically that the injury that is analyzed is demyelination. Regeneration is now days a complex issue, the SVZ may respond differently to different types of injuries and therefore the title should be more specific.

We thank the reviewer for this suggestion. The title is now changed accordingly to “Non-Canonical Wnt Signaling Regulates Neural Stem Cell Quiescence During Homeostasis and After Demyelination”.

Minor points:

- Legend of Figure 1: correct “atleast” (twice). In legend of Figure 3: correct “leftmost”.

We apologize for this mistake and have made the appropriate corrections.

- What is SCWM in Methods (“Lysolecithin induced demyelination paradigm”)?

We now added the full term “subcortical white matter” for the abbreviation (SCWM) in the methods.

- In the panels depicting in situ hybridization for different wnt elements shown in Figure S1a is very difficult to interpret what is a signal. Furthermore, it is not appropriate to include photographs taken from the Allen brain atlas as part of the results, even if they are supplementary data.

We apologize for this issue. We removed this data from the revised manuscript.

- In the legend of Figure S1, correct the sentence “error bars represent mean+SEM”.

The legend has been corrected accordingly.

- Figure S2a is not quantitated and, although it is mentioned in the text, it is unclear what it represents.

We now added quantifications to these blots (Supplementary Figure 2a-b). This data suggests that Wnt5a treatment/stimulation induces Notch1 expression, but does not impact the levels of Notch1 ligands Delta like 1 (Dll1) and Jagged 1 (Jag1). This data suggests that Cdc42 activity has a direct role in regulating only Notch1 receptor expression but not its ligands.

- Line 197: is Fig. 3c not 2h.

We apologize for this mistake and have corrected this figure citation.

- Improve labels in the flow cytometry plots.

We improved the resolution of labels on all of the pertinent flow cytometry plots.

- Define XAV939 somewhere (Figure S2).

We now defined XAV939, a pharmacological tankyrase inhibitor that antagonizes β -catenin dependent Wnt signaling.

- The authors could add the labels “canonical wnt ligands” and “non-canonical wnt ligands” to mRNA graphs in Figure 4.

We have now added these labels in the revised manuscript accordingly.

- Line 387, please correct “post transnationally”.

We apologize for this mistake and have corrected this issue.

- Line 410, show the same number of decimals in the data.

We have now corrected this issue.

Reviewer #3 (Remarks to the Author):

In this study, Chavali et al. report that the quiescent state of neural stem cells is maintained by a non-canonical Wnt signaling, through the activation of cdc42. Interestingly, after a demyelinating insult, this non-canonical Wnt signaling is decreased, and the authors infer that this is required to promote the differentiation of stem cells for repair. There are several interesting findings in this paper. Although Wnt pathways components have been described in stem cells, the distinction between canonical and non-canonical Wnt signaling is new, and adds a complexity to the regulation of quiescence in neural stem cells. This is a large and broad manuscript, and some of the conclusions are not clearly documented. The cellular localization of the many molecular changes is not always clear, for example. Also, at several points the authors do not clearly differentiate between quiescent stem cells and activated stem cells. In addition, the authors should make clear some of the quantitative analyses of cell numbers.

Specific Points:

1. “NSCs” were separated out as GFP+ in the GFAP:GFP mouse. This population should include both quiescent and activated stem cells. Activated stem cells are thought to differ from the quiescent by their expression of EGFR. Were these two populations separated to look at the expression of Wnt pathway genes or to look at the effects of Wnt signaling?

We thank the reviewer for this observation. While we did not separate these populations to look at the expression of Wnt pathway genes, the RNA sequencing data from Codega et al.,²³ in which the quiescent and activated NSCs were prospectively isolated based on their EGFR expression demonstrates that the canonical Wnt signaling pathway downstream targets and effectors are highly enriched in the activated NSCs (GFAP+CD133+EGFR+) and the Cdc42 dependent non-canonical Wnt pathway effectors and β -catenin pathway inhibitors are significantly enriched in the quiescent NSCs (GFAP+CD133+EGFR-)^{18, 30-33}. We are presenting some of this data for your reference (Reviewer Figure 5a-b). In addition, our

immunohistochemical analysis of adult SVZ for GFAP, Nestin and Axin2 revealed that quiescent NSCs (GFAP+Nestin- cells) did not express Axin2 (a downstream canonical Wnt signaling target; Reviewer Figure 5c, indicated by arrowheads), while the GFAP+Nestin+ activated NSCs²³ displayed Axin2 expression (Reviewer Figure 5c, indicated by arrows).

Reviewer Figure 5: Expression of Canonical and Non-Canonical Wnt signaling effectors in activated and quiescent NSCs of SVZ. a) RNA expression data of canonical Wnt signaling pathway downstream targets and effector genes are highly enriched in activated NSCs. b) RNA expression of non-canonical Wnt and Cdc42 pathway effectors are enriched in quiescent NSCs. c) Adult mouse brain sections stained for GFAP (green), Nestin (blue), and Axin2 (red) reveal that quiescent NSCs (arrowhead) do not, but activated NSCs (arrows) do express Axin2.

2. The SVZ also contains astrocytes. How were these factored into the various counting paradigms in the GFAP:GFP mice?

We agree with the reviewer that the GFAP::GFP+ population in the SVZ also contains astrocytes. However, the counts from these reporter mice were only performed on the most superficial layer of the SVZ from the wholemounts, considering only apical GFAP::GFP+ cells, which are known to be quiescent NSCs^{11, 34, 35}. For the GFAP::GFP sorting experiments (Supplementary Figure 1b), we only used the high intensity GFP cells that are considered to be quiescent NSCs^{23, 36}.

3. In Figure 1, were the NSCs the entire GFP+ population from the SVZ?

We have now reorganized this data. Based on reviewer # 2's comments we removed the NSC data presented in the earlier version. In the revised Figure 1, we now only present data on proteomic analysis of the SVZ.

4. For Wnt signaling molecules the whole SVZ was studied, not separate populations. We don't know what cells in the SVZ express the various Wnt pathway components. For

example, is Cdc42 expression confined to quiescent stem cells or is its expression also seen in the more mitotically active stem cells, or even in further derivatives?

As shown above (Reviewer Figure 5), the non-canonical Wnt signaling and its downstream targets and effectors are enriched in quiescent NSCs²³, while the canonical Wnt pathway and its downstream targets are highly enriched in activated NSCs²³. In addition, we also report that Wnt3a does not impact the apical GFAP::GFP+ cell numbers (considered as quiescent NSCs^{34, 35}; Supplementary Figure 3d, f).

5. The “polarization” promoted by Wnt5a and shown in Figure 3c is not at all clear. In Figure 3c, why does the control Cdc42 merged with tubulin show cell processes, yet the GM130 merged with tubulin does not? It’s difficult to see the difference between GM130 localization in controls and Wnt52. Both have a paranuclear localization.

We have now revised this data regarding the cell polarization mediated by Wnt5a. Figure 3c now shows that Wnt5a promotes Cdc42 and alpha tubulin polarization, while addition of CASIN prevents this effect. To appreciate this polarization better, we now provide intensity distribution plots for both Cdc42 and tubulin (Figure 3c; bottom panels). Furthermore, we also performed immunocytochemistry with anti-N-cadherin antibodies on NSCs after Wnt5a stimulation, consistent with our idea Wnt5a treated NSCs displayed a polarized morphology as compared to control, and this effect was reduced in the presence of CASIN (Figure 3d).

6. Wnt5a applied to SVZ explants induced expression of Notch1, Id1, and N-cadherin. The Id1 Western blot does not show much of a difference between Control and Wnt5a, however. A quantitation would be important here.

To improve this data, we are now providing the quantifications for Notch1 (Supplementary Figure 2a-b) and N-Cadherin (Figure 3e-f) immunoblots. For the Id1 immunoblots: the Wnt5a+CASIN condition did not show a band at low exposure times to obtain appropriate densitometry quantifications and higher exposure times resulted in signal saturation in the Control and Wnt5a treated conditions, as the reviewer rightly pointed out. We therefore removed the Id1 immunoblot in Figure 3. However, *Id1-3* mRNA levels after Wnt5a stimulation are presented in Figure 3g and this data shows that Wnt5a induces an upregulation of *Id1* mRNA levels in a Cdc42 dependent manner.

7. For the BrdU quantification in Figure 3, how many sections per mouse were examined to determine labeling index? Does this result suggest that the GFAP+/Sox3+ cells are the only ones that activate Cdc42 in response to Wnt5a?

In this set of experiments, we quantified the number of BrdU label retaining cells from at least 4-6 sections from each animal and 4 animals in total for each condition. It must be noted that both GFAP and Sox2 were only used as the markers NSC markers. In this revised version, we are excluding these markers for the quantification purposes and are only presenting the numbers of BrdU label retaining cells (per Reveiwer#1’s suggestion) in conjunction with the ki67 counts in all the experiments as other reviewers suggested.

As to which cells respond to Wnt5a and activate Cdc42, we believe that the quiescent NSCs (GFAP+CD133+EGFR- cells) have the appropriate gene expression pattern to both respond and allow for further downstream signaling to induce the expression of Notch1, Id1 and N-Cadherin (as shown in reviewer figure 5). However, future studies will be required to identify other effectors of Wnt5a/Cdc42 signaling axis in the SVZ NSCs.

8. The authors note that Cdc42 keeps cortical progenitors in cycle in an EGF dependent manner (line 238). I don't quite understand the point of this comparison, since quiescent NSCs don't express EGFR.

We apologize for this confusion. We removed this sentence from the revised version of the manuscript.

9. The authors postulate that blocking Cdc42 with CASIN results in activation of quiescent NSCs and therefore fewer BrdU+ quiescent NSCs. They find fewer BrdU+ cells. Another interpretation of this finding, however, is that blocking Cdc42 interferes with mitosis or cytokinesis (see Sakemori et al., JCI 2012 122:1052, who find that CASIN blocks cytokinesis of intestinal stem cells).

We appreciate this observation. To rule out the possibility that CASIN blocks the cytokinesis, we performed ki67 immunostaining on brains obtained from the CASIN treated mice, and our analysis showed an increase in the total numbers of ki67+ cells in the SVZ in addition to decreased BrdU label retaining cells after CASIN treatment. This data is now presented in the revised manuscript in supplementary figure 2c.

10. Do the authors have independent evidence that CASIN effectively blocks Cdc42 activity in this system?

We are now providing the quantified immunoblot results which show that Cdc42-GTP levels are significantly decreased in the SVZ following *in vivo* CASIN treatment (Figure 3h-k), suggesting that CASIN effectively blocks Cdc42 activity.

11. In the experiments in which SVZs were infected with AAVs, why was the effect of Wnt3a the same as that of a LacZ virus?

We apologize for not elaborating these results. As mentioned above in point 1 and as shown in revised supplementary figure 3d-f, Wnt3a does not impact the quiescent NSCs. This could possibly be due to the expression of Dkk3, a canonical Wnt signaling inhibitor in quiescent NSCs²³ or that they do not express the receptor complex that Wnt3a can bind to and activate a downstream signal. We also show that Wnt3a induces proliferation in the SVZ (Supplementary Figure 3c, d-e). Therefore, Wnt3a overexpression in the SVZ during a demyelinating injury doesn't rescue the BrdU label retaining quiescent NSCs and so its effect remains the same as a control vector (LacZ).

12. Cdc42 induces the expression of a variety of adhesion molecules, although the authors provide no evidence that any of them functions to anchor NSCs to any particular cell type or matrix in the SVZ. The authors should not conclude definitively that these molecules are involved in anchorage. In a similar vein, the decrease of JAM-C and cadherins after a demyelinating insult is interesting, but it's not known in what cell(s) this decrease occurs.

We thank the reviewer for this observation and would like to clarify this point further. We show that Wnt5a mediated Cdc42 activity mainly effects the expression of N-Cadherin and Id1 in the SVZ (Figure 3d, g). In addition, we show that at the peak of a demyelination injury (7DPI) when Cdc42 activity is decreased, expression of JAM-C, N-Cadherin, E-Cadherin and Id1 are also decreased in the GFAP+ NSCs lining the ventricular wall in the SVZ (Supplementary Figure 5).

Although we do not show these adhesion molecules maintain the NSC anchorage to the SVZ niche in this work, previous reports have shown the functional role these molecules play in NSC adhesion. In this context, Porlan et al²⁵, observed that N-Cadherin expression in the NSCs is required for the NSC and ependymal cell contact and its relevance in maintaining quiescence. Similarly, Id proteins have been reported to be required for the anchorage of NSCs to their niche and that the disruption in their function results in loss of adhesion to the SVZ niche³⁷. Furthermore, Jam-C and E-Cadherin have also been reported to maintain the adhesion of NSCs to the ventricular wall^{38,39}.

References:

1. Piccin, D. & Morshead, C.M. Wnt signaling regulates symmetry of division of neural stem cells in the adult brain and in response to injury. *Stem cells* **29**, 528-538 (2011).
2. Kippin, T.E., Martens, D.J. & van der Kooy, D. p21 loss compromises the relative quiescence of forebrain stem cell proliferation leading to exhaustion of their proliferation capacity. *Genes & development* **19**, 756-767 (2005).
3. Morshead, C.M., Craig, C.G. & van der Kooy, D. In vivo clonal analyses reveal the properties of endogenous neural stem cell proliferation in the adult mammalian forebrain. *Development* **125**, 2251-2261 (1998).
4. Aguirre, A., Rubio, M.E. & Gallo, V. Notch and EGFR pathway interaction regulates neural stem cell number and self-renewal. *Nature* **467**, 323-327 (2010).
5. Daynac, M. et al. Hedgehog Controls Quiescence and Activation of Neural Stem Cells in the Adult Ventricular-Subventricular Zone. *Stem Cell Reports* **7**, 735-748 (2016).
6. Kersey, P.J. et al. The International Protein Index: an integrated database for proteomics experiments. *Proteomics* **4**, 1985-1988 (2004).
7. Elias, J.E. & Gygi, S.P. Target-decoy search strategy for mass spectrometry-based proteomics. *Methods Mol Biol* **604**, 55-71 (2010).
8. Tabb, D.L., McDonald, W.H. & Yates, J.R., 3rd DTASelect and Contrast: tools for assembling and comparing protein identifications from shotgun proteomics. *J Proteome Res* **1**, 21-26 (2002).
9. Carvalho, P.C., Hewel, J., Barbosa, V.C. & Yates, J.R., 3rd Identifying differences in protein expression levels by spectral counting and feature selection. *Genet Mol Res* **7**, 342-356 (2008).
10. Liu, H., Sadygov, R.G. & Yates, J.R., 3rd A model for random sampling and estimation of relative protein abundance in shotgun proteomics. *Anal Chem* **76**, 4193-4201 (2004).
11. Mirzadeh, Z., Merkle, F.T., Soriano-Navarro, M., Garcia-Verdugo, J.M. & Alvarez-Buylla, A. Neural stem cells confer unique pinwheel architecture to the ventricular surface in neurogenic regions of the adult brain. *Cell stem cell* **3**, 265-278 (2008).
12. Cheung, T.H. & Rando, T.A. Molecular regulation of stem cell quiescence. *Nat Rev Mol Cell Biol* **14**, 329-340 (2013).
13. Fabregat, A. et al. The Reactome pathway Knowledgebase. *Nucleic acids research* **44**, D481-487 (2016).

14. Szklarczyk, D. *et al.* The STRING database in 2017: quality-controlled protein-protein association networks, made broadly accessible. *Nucleic acids research* **45**, D362-D368 (2017).
15. Penzo-Mendez, A., Umbhauer, M., Djiane, A., Boucaut, J.C. & Riou, J.F. Activation of Gbetagamma signaling downstream of Wnt-11/Xfz7 regulates Cdc42 activity during *Xenopus* gastrulation. *Developmental biology* **257**, 302-314 (2003).
16. Florian, M.C. *et al.* A canonical to non-canonical Wnt signalling switch in haematopoietic stem-cell ageing. *Nature* **503**, 392-396 (2013).
17. Joberty, G., Petersen, C., Gao, L. & Macara, I.G. The cell-polarity protein Par6 links Par3 and atypical protein kinase C to Cdc42. *Nature cell biology* **2**, 531-539 (2000).
18. Sugimura, R. *et al.* Noncanonical Wnt signaling maintains hematopoietic stem cells in the niche. *Cell* **150**, 351-365 (2012).
19. Yang, L. *et al.* Rho GTPase Cdc42 coordinates hematopoietic stem cell quiescence and niche interaction in the bone marrow. *Proceedings of the National Academy of Sciences of the United States of America* **104**, 5091-5096 (2007).
20. Choi, S.C. & Han, J.K. *Xenopus* Cdc42 regulates convergent extension movements during gastrulation through Wnt/Ca²⁺ signaling pathway. *Developmental biology* **244**, 342-357 (2002).
21. Fancy, S.P. *et al.* Dysregulation of the Wnt pathway inhibits timely myelination and remyelination in the mammalian CNS. *Genes & development* **23**, 1571-1585 (2009).
22. Yuen, T.J. *et al.* Oligodendrocyte-encoded HIF function couples postnatal myelination and white matter angiogenesis. *Cell* **158**, 383-396 (2014).
23. Codega, P. *et al.* Prospective identification and purification of quiescent adult neural stem cells from their in vivo niche. *Neuron* **82**, 545-559 (2014).
24. Clevers, H., Loh, K.M. & Nusse, R. Stem cell signaling. An integral program for tissue renewal and regeneration: Wnt signaling and stem cell control. *Science* **346**, 1248012 (2014).
25. Porlan, E. *et al.* MT5-MMP regulates adult neural stem cell functional quiescence through the cleavage of N-cadherin. *Nature cell biology* **16**, 629-638 (2014).
26. van Noort, M., Meeldijk, J., van der Zee, R., Destree, O. & Clevers, H. Wnt signaling controls the phosphorylation status of beta-catenin. *The Journal of biological chemistry* **277**, 17901-17905 (2002).
27. Wang, Z., Shu, W., Lu, M.M. & Morrisey, E.E. Wnt7b activates canonical signaling in epithelial and vascular smooth muscle cells through interactions with Fzd1, Fzd10, and LRP5. *Mol Cell Biol* **25**, 5022-5030 (2005).
28. Bisson, J.A., Mills, B., Paul Helt, J.C., Zwaka, T.P. & Cohen, E.D. Wnt5a and Wnt11 inhibit the canonical Wnt pathway and promote cardiac progenitor development via the Caspase-dependent degradation of AKT. *Developmental biology* **398**, 80-96 (2015).
29. Mikels, A.J. & Nusse, R. Purified Wnt5a protein activates or inhibits beta-catenin-TCF signaling depending on receptor context. *PLoS Biol* **4**, e115 (2006).
30. Bernatik, O. *et al.* Functional analysis of dishevelled-3 phosphorylation identifies distinct mechanisms driven by casein kinase 1 and frizzled5. *The Journal of biological chemistry* **289**, 23520-23533 (2014).

31. Smalley, M.J. *et al.* Dishevelled (Dvl-2) activates canonical Wnt signalling in the absence of cytoplasmic puncta. *J Cell Sci* **118**, 5279-5289 (2005).
32. Moon, R.T. Wnt/beta-catenin pathway. *Sci STKE* **2005**, cm1 (2005).
33. Sugimura, R. & Li, L. Noncanonical Wnt signaling in vertebrate development, stem cells, and diseases. *Birth Defects Res C Embryo Today* **90**, 243-256 (2010).
34. Kokovay, E. *et al.* VCAM1 is essential to maintain the structure of the SVZ niche and acts as an environmental sensor to regulate SVZ lineage progression. *Cell stem cell* **11**, 220-230 (2012).
35. Kokovay, E. *et al.* Adult SVZ lineage cells home to and leave the vascular niche via differential responses to SDF1/CXCR4 signaling. *Cell stem cell* **7**, 163-173 (2010).
36. Dulken, B.W., Leeman, D.S., Boutet, S.C., Hebestreit, K. & Brunet, A. Single-Cell Transcriptomic Analysis Defines Heterogeneity and Transcriptional Dynamics in the Adult Neural Stem Cell Lineage. *Cell reports* **18**, 777-790 (2017).
37. Niola, F. *et al.* Id proteins synchronize stemness and anchorage to the niche of neural stem cells. *Nature cell biology* **14**, 477-487 (2012).
38. Stelzer, S. *et al.* JAM-C is an apical surface marker for neural stem cells. *Stem Cells Dev* **21**, 757-766 (2012).
39. Kuo, C.T. *et al.* Postnatal deletion of Numb/Numbl like reveals repair and remodeling capacity in the subventricular neurogenic niche. *Cell* **127**, 1253-1264 (2006).

Reviewers' comments:

Reviewer #1 (Remarks to the Author):

The authors have engaged in an extensive revision of their original manuscript. They provide important additional data, including quantifications of western blots and key additional experiment with Wnt5a overexpression following demyelination.

Due to this extensive revision, the manuscript is particularly lengthy. Its reviewing would have been greatly facilitated if authors had underlined parts of the manuscript that have been modified. Also numbering of the pages would have been appreciated.

Most of my original concerns have been appropriately addressed. However, one major concern remains. It is related to the label retaining protocol and quantifications, which I considered to be sub-optimal owing to the small number of cells quantified. Authors have answered these concerns by providing absolute numbers of BrdU label retaining cells that were counted in the entire length of the SVZ in each section, in all experiments. As shown in fig 3m, 4j, 6d... these numbers are extremely small (8 to 10 cells in control animals). I insist that performing quantifications on such low number of cells is cumbersome and may lead to false conclusions. I understand that addressing this concern is not straightforward as it would require repeating a large part of this work, but I believe it to be a valid concern that has not been appropriately addressed.

These quantifications are further complicated by the "activation" and BrdU dilution in the LRCs following demyelination injury. In supplementary figure 3a-c (wrongly referred as suppl Fig 4a-c in the result section), authors indeed show an increase in the number of diluted BrdU cells co-expressing PCNA, whereas a general decrease in the number of LRCs was observed at the same time (average of 5 LRCs/SVZ). It is unclear if these two populations of cells were quantified separately (nor on which criteria), or if the second population represents a fraction of the first population (and therefore an even smaller number of cells). This raises additional concerns on the reliability and accuracy of the histology/quantifications performed in this study.

Authors attempt at circumventing this concern by using FAC-sorting to quantify quiescent vs activated stem cells, as illustrated in the rebuttal letter. This is a valid approach that could indeed substantiate the suboptimal histological analysis. These experiments should however be performed for all experiments where LRC are quantified and included in the manuscript, which I agree, represents however a large amount of additional work.

Other concerns:

Results, first paragraph : it is written « Using these proteins, we performed gene ontology analysis and detected alterations in GO categories for cell adhesion, cytoskeletal remodeling, and cellular response to stimulus and metabolic activity, all of which are indicative of NSC activation (Fig. 1a-b) ». This is not supported by the data, as the "cytoskeletal remodeling" GO category does not appear in the figure.

Fig1: Panel 1d is interesting as it further highlight Wnt/Rho-GTPase pathways interactions in the SVZ following demyelination. It is however not readable.

To test if this decrease in Notch signaling had a direct impact on the self-renewal capacity of NSCs, authors have performed in vitro neurosphere assays. In these experiments an equal number of GFAP::GFP+ cells were plated. They show a decrease in the number of neurospheres formed. This is puzzling as one would expect an increase if NSCs were indeed activated.

The title "Forced Induction of Non-Canonical Wnt Signaling by Wnt5a In the SVZ is Sufficient to Rescue Injury Induced Activation of Quiescent NSCs" is confusing, as Wnt5a block the activation of quiescent NSCs induced by demyelination. Please reformulate.

Suppl Figure 6: LacZ should appear nuclear which is not the case on several of the panels. Also, the figure suggest that LacZ is used as a reporter for all AAV. This is not described in the material and methods. Please clarify.

Authors have added Wnt5a overexpression (w/wo casin treatment) following demyelination. These are difficult experiments, which definitely add to the manuscript. I believe that with these new results, NICD overexpression experiments could be removed from the manuscript in order to simplify it (figure 7).

Reviewer #2 (Remarks to the Author):

The authors have done a very good job in addressing all my concerns. In particular, they have more comprehensively analyzed the proteomics data to more substantially support the driving hypothesis of the work, have provided important controls for some of the experiments, and have improved the in vivo analyses. The latter was especially relevant as one of the most interesting aspects of this manuscript in the extent of the in vivo approaches. However, I would still like to suggest a few changes that could improve the experimental support for the conclusions and the readability of the manuscript.

One of my concerns with some of the in vivo analyses was that dilution of BrdU alone is an imperfect read-out for the potential activation of NSCs. I had suggested in my review that they could approach the problem as they had done for the analysis of lysolecithin-treated mice shown in Figure 4 in which they had analyzed PCNA-immunopositivity in the LRCs. In this new version, the authors have added stainings with antibodies to Ki67 and this is valuable for the interpretation of the results, but they are providing data on Ki67+ cell numbers independently of the BrdU labeling. I still think that it would be desirable to have some assessment of activation in the BrdU-LRC themselves. If not, at least they should show the data on Ki67+ cells and in PCNA+ cells together with the data on BrdU-LRCs (as they do in Figure 6) and not in supplementary information as these are highly relevant for the conclusions. If it is a matter of space, my humble opinion is that Hes transcripts of Figure 5c are less relevant for the main flow of the story.

The authors need to make an effort in conveying the information in their figures in a way that is easier to follow. Some examples of things that could be changed to facilitate the reading of the manuscript are listed below:

- Graphs in Figure 2 are labeled in different, and therefore confusing, ways. Compare "i" and "k". Both Y axis could be labeled "relative protein levels", putting the name of the protein in X and using the legend for the treatments to standardize.

- In the in vivo analyses of CASIN treated-mice shown in Figure 3 the authors change from control situation (CTRL) to "vehicle". That is OK but in Figure 4, which shows the experiments with lysolecithin, the authors switch to "NaCl" to refer to the vehicle solution. Homogeneity would be desirable.

- Average numbers of LRCs found in several graphs are not "per SVZ" but "per SVZ section".

- LRCs are labeled as such in the graphs but in the text the nomenclature has been changed to LR-qNSC. I would propose keeping the abbreviation LRC. Indeed, in some parts of the manuscript the authors use the term dormant NSCs as slowly dividing. It is now considered that dormant NSCs cannot be labeled with BrdU and the term qNSC refers to this type of cells to distinguish them from the activated slowly-dividing aNSCs.

- In Figure 6, the pictures for the vehicle and Lyso treatments are from different hemispheres and, therefore, it makes sense that the lateral ventricle in the photographs is either to the left or to the right of the SVZ tissue. But it is very confusing finding the same situation in Figure 5, before the experimental scheme. And, even more, that the lateral ventricles are inverted with respect to the panels in Figure 6.

Reviewer #3 (Remarks to the Author):

Authors have answered all main points and improved the paper substantially.

Reviewer # 1

The authors have engaged in an extensive revision of their original manuscript. They provide important additional data, including quantifications of western blots and key additional experiment with Wnt5a overexpression following demyelination.

Due to this extensive revision, the manuscript is particularly lengthy. Its reviewing would have been greatly facilitated if authors had underlined parts of the manuscript that have been modified. Also, numbering of the pages would have been appreciated.

We would like to thank the reviewer for their continued interest in our work. We also apologize for the issue of not highlighting the changes made in the revised version. Since major revisions were done, both in terms of experiments and in writing up the results, discussion, and methods sections, we omitted highlighting all the individual changes. However, in this current version, we have underlined the changes done in the manuscript from the original submission and included the page numbers, as per the reviewer's request.

Most of my original concerns have been appropriately addressed. However, one major concern remains. It is related to the label retaining protocol and quantifications, which I considered to be sub-optimal owing to the small number of cells quantified. Authors have answered these concerns by providing absolute numbers of BrdU label retaining cells that were counted in the entire length of the SVZ in each section, in all experiments. As shown in fig 3m, 4j, 6d... these numbers are extremely small (8 to 10 cells in control animals). I insist that performing quantifications on such low number of cells is cumbersome and may lead to false conclusions. I understand that addressing this concern is not straightforward as it would require repeating a large part of this work, but I believe it to be a valid concern that has not been appropriately addressed.

We appreciate this concern, and want to take this opportunity to clarify this issue further. In the previous version of our manuscript, we inadvertently mislabeled the axes in the LRC experiments as "Avg. number of LRCs/SVZ" instead of "Avg. number of LRCs/Section", which may have been misinterpreted as fewer numbers of cells analyzed. We have now corrected the axes labels to reflect this. We would also like to again highlight that we analyzed ~4-8 coronal sections (1.045mm to -0.08mm from bregma) from each of the control and experimental conditions, and the total number of cells quantified from n=4-6 animals were ~200-300 cells in figures 3m, 4j, and 6d. In this new revised manuscript, we are presenting these total numbers of cells quantified for each condition for all the experiments in the pertaining figure legends. We are also providing details on cell counting and analysis in the methods sections.

In addition, we are also including Reviewer Figure 1 (see below), in which we are depicting a typical confocal projection of the SVZ from a coronal section, immunostained for Ki67 and BrdU from control and lysolecithin injected hemispheres from 7DPI brains. In this image, it can be appreciated that the control (contralateral) SVZ has about ~8 BrdU+ label retaining cells in a 30µm section, and this number decreases to ~3 cells in the SVZ of the corresponding lysolecithin injected (ipsilateral) hemisphere.

Furthermore, we would also like to point out that our cell quantifications are in agreement with recent reports that have described similar numbers of label retaining cells in the SVZ¹⁻⁵ using similar BrdU labeling paradigms. Additionally, a recent study employing doxycycline inducible histone 2b(H2B)-GFP expression mice also reported comparable numbers of quiescent NSCs/section⁶.

Reviewer Figure 1: Confocal Projection of V-SVZ from control and lysolecithin injected hemispheres at 7DPI, immunostained for Ki67 and BrdU. It can be observed that control hemisphere SVZ has higher numbers of BrdU-LRCs compared to the lysolecithin injected hemisphere SVZ (indicated by arrows). Arrowheads show Ki67+BrdU+ activated cells. Scale bar represents 50µm.

These quantifications are further complicated by the “activation” and BrdU dilution in the LRCs following demyelination injury. In supplementary figure 3a-c (wrongly referred as suppl Fig 4a-c in the result section), authors indeed show an increase in the number of diluted BrdU cells co-expressing PCNA, whereas a general decrease in the number of LRCs was observed at the same time (average of 5 LRCs/SVZ). It is unclear if these two populations of cells were

quantified separately (nor on which criteria), or if the second population represents a fraction of the first population (and therefore an even smaller number of cells). This raises additional concerns on the reliability and accuracy of the histology/quantifications performed in this study.

First, we would like to clarify that the figure numbers were referred to correctly in the revised version (The LRC-BrdU and PCNA co-labeling experiments were a part of supplementary figure 4 in the revised version; we now moved this dataset to Figure 5 as requested by reviewer #2).

The diluted BrdU+ cells were not quantified as part of LRC population, as they are no longer functionally quiescent, which is determined by expression of PCNA (see also Reviewer Figure 2). During a demyelination injury BrdU-label retaining cells in the SVZ enter the cell cycle and can be detected by their PCNA expression, and as these LRCs divide, there is dilution of the BrdU label. To explain this in a quantitative manner, we plotted the BrdU intensity of both PCNA+ (activated) and PCNA- (quiescent) cells (Figure 5c). As it can be observed in Figure 5a-c the BrdU+PCNA+ activated cells have reduced BrdU intensity.

It must be also noted that the label retaining quiescent NSCs in the SVZ niche are a very rare population constituting only about ~2.5% of the total SVZ cells⁷⁻⁹. Finally, as mentioned above, we have quantified over ~200-300 cells (BrdU+ LRCs) from n=4-6 animals to confirm the reliability of our LRC quantifications, and as mentioned earlier these numbers are in agreement with previous reports that have used the BrdU labeling paradigm we employed here or similar approaches¹⁻⁶.

Authors attempt at circumventing this concern by using FAC-sorting to quantify quiescent vs activated stem cells, as illustrated in the rebuttal letter. This is a valid approach that could indeed substantiate the suboptimal histological analysis. These experiments should however be performed for all experiments where LRC are quantified and included in the manuscript, which I agree, represents however a large amount of additional work.

We appreciate the reviewer's remarks regarding the FAC-sorting approach. The FAC-sorting data provided in the earlier rebuttal letter was to provide additional evidence for the activation of NSCs and the subsequent decrease in the quiescent NSC population after a demyelination injury. While this analysis provides additional validation of our data, we strongly believe that the thymidine analog BrdU-label retention approach is the best way to functionally characterize and quantitatively assess the quiescence of NSCs¹⁰⁻¹⁴. As shown here and by other groups¹⁻⁵, the BrdU-label retaining assay can assertively demonstrate quiescence/activation and cell-cycle entry of rare NSCs *in vivo* using immunohistochemistry and confocal microscopy. Furthermore, although it has not been conclusively demonstrated yet, there is a possibility of NSCs dynamically shuttling between quiescence and activation^{8,15} which we believe will further complicate the use of the FAC-sorting strategy to conclusively determine the numbers of quiescent NSCs in their native niche.

Other concerns:

Results, first paragraph: it is written « Using these proteins, we performed gene ontology analysis and detected alterations in GO categories for cell adhesion, cytoskeletal remodeling, and cellular response to stimulus and metabolic activity, all of which are indicative of NSC activation (Fig. 1a-b) ». This is not supported by the data, as the “cytoskeletal remodeling” GO category does not appear in the figure.

We apologize for this issue. The “cytoskeletal remodeling” category is now removed from the text.

Fig1: Panel 1d is interesting as it further highlight Wnt/Rho-GTPase pathways interactions in the SVZ following demyelination. It is however not readable.

We thank the reviewer for this observation. We have now changed this figure to make it more legible.

To test if this decrease in Notch signaling had a direct impact on the self-renewal capacity of NSCs, authors have performed in vitro neurosphere assays. In these experiments an equal number of GFAP::GFP+ cells were plated. They show a decrease in the number of neurospheres formed. This is puzzling as one would expect an increase if NSCs were indeed activated.

We thank the reviewer for this observation and we would like to clarify this point further. It must be noted that the neurosphere formation assay was performed using FAC-sorted GFAP::GFP+ cells in NSC culture media (DMEM/F12 basal media) in the presence of only bFGF. The decrease in the numbers of secondary neurospheres indicated a decrease in self-renewal and an increase in the lineage progression, which was further confirmed by an *in vivo* increase in the GFAP⁺EGFR⁺ population (Figure 5h-i). The absence of EGF mitogen in the neurosphere media explains the reason for not observing an increase in the numbers of secondary neurospheres. Indeed, we have previously reported an increase in the neurospheres numbers obtained from the SVZ cells after a demyelination injury that were maintained in both EGF and bFGF ¹⁶.

The title “Forced Induction of Non-Canonical Wnt Signaling by Wnt5a In the SVZ is Sufficient to Rescue Injury Induced Activation of Quiescent NSCs” is confusing, as Wnt5a block the activation of quiescent NSCs induced by demyelination. Please reformulate.

We have now changed the title describing this part of the results to “Forced Expression of Wnt5a in the SVZ Inhibits the Demyelination Induced Activation of Quiescent NSCs.”

Suppl Figure 6: LacZ should appear nuclear which is not the case on several of the panels. Also, the figure suggest that LacZ is used as a reporter for all AAV. This is not described in the material and methods. Please clarify.

We apologize for this technical issue. In the revised manuscript, we are now providing new images that show a nuclear staining pattern of LacZ. We have also updated the methods to reflect that LacZ was used as a reporter in all AAV infections.

Authors have added Wnt5a overexpression (w/wo casin treatment) following demyelination. These are difficult experiments, which definitely add to the manuscript. I believe that with these new results, NICD overexpression experiments could be removed from the manuscript in order to simplify it (figure 7).

We thank the reviewer for this comment. We believe these additional *in vivo* experiments requested earlier by the reviewers have strengthened the work. As per reviewer’s suggestion to simplify the data presented in the Figure 7, we are now moving the Notch overexpression data to Supplementary Figure 7.

Reviewer # 2

The authors have done a very good job in addressing all my concerns. In particular, they have more comprehensively analyzed the proteomics data to more substantially support the driving hypothesis of the work, have provided important controls for some of the experiments, and have improved the in vivo analyses. The latter was especially relevant as one of the most interesting aspects of this manuscript in the extent of the in vivo approaches. However, I would still like to suggest a few changes that could improve the experimental support for the conclusions and the readability of the manuscript.

We would like to thank the reviewer for this point and appreciate their comment on highlighting the importance of our *in vivo* approaches.

One of my concerns with some of the *in vivo* analyses was that dilution of BrdU alone is an imperfect read-out for the potential activation of NSCs. I had suggested in my review that they could approach the problem as they had done for the analysis of lysolecithin-treated mice shown in Figure 4 in which they had analyzed PCNA-immunopositivity in the LRCs. In this new version, the authors have added stainings with antibodies to Ki67 and this is valuable for the interpretation of the results, but they are providing data on Ki67+ cell numbers independently of the BrdU labeling. I still think that it would be desirable to have some assessment of activation in the BrdU-LRC themselves. If not, at least they should show the data on Ki67+ cells and in PCNA+ cells together with the data on BrdU-LRCs (as they do in Figure 6) and not in supplementary information as these are highly relevant for the conclusions. If it is a matter of space, my humble opinion is that Hes transcripts of Figure 5c are less relevant for the main flow of the story.

This is a well-considered point. Indeed, we performed the ki67 staining in conjunction with the BrdU labeling (please see reviewer figure 1). The reason for not providing the quantitative data on the double positive cells (ki67+BrdU LRC+ cells) is that in the control conditions virtually none of the BrdU-LRCs co-expressed ki67 or PCNA (see Reviewer Figure 1 and 2 (in page number 6 here)). In the reviewer figure 2 we are also presenting additional confocal images of the dorsal, lateral, and ventral SVZ regions, where this can be observed more clearly. Therefore, quantification and comparison of the double positive cells becomes challenging. But as per the reviewer's suggestion we now present the BrdU/PCNA dataset in the main figures (Figure 5).

The authors need to make an effort in conveying the information in their figures in a way that is easier to follow. Some examples of things that could be changed to facilitate the reading of the manuscript are listed below:

- Graphs in Figure 2 are labeled in different, and therefore confusing, ways. Compare “i” and “k”. Both Y axis could be labeled “relative protein levels”, putting the name of the protein in X and using the legend for the treatments to standardize.

We believe the reviewer was referring to Figure 3i and 3k. We keep the labeling of axes in figure 3i the same as the rest of the graphs in figure 3 and 4 as it is a quantification of Cdc42-GTP levels, which we consistently labeled as Cdc42-GTP/Total Cdc42 as per reviewer #1's suggestion during the earlier revisions. We have now changed the graph in figure 3k as per the reviewer's suggestion (Y Axis: “protein levels” and X axis: “name of the protein”).

- In the *in vivo* analyses of CASIN treated-mice shown in Figure 3 the authors change from control situation (CTRL) to “vehicle”. That is OK but in Figure 4, which shows the experiments with lysolecithin, the authors switch to “NaCl” to refer to the vehicle solution. Homogeneity would be desirable.

We thank the reviewer for this observation. For consistency purposes, we have now changed Vehicle and NaCl to “CTRL” in all the figures and graphs, and for easier understanding this is reflected in all the pertaining cartoon depictions.

- Average numbers of LRCs found in several graphs are not “per SVZ” but “per SVZ section”.

We apologize for this inadvertent mistake. The LRC quantifications are indeed per SVZ section, we have now revised all the figures accordingly.

- LRCs are labeled as such in the graphs but in the text the nomenclature has been changed to LR-qNSC. I would propose keeping the abbreviation LRC. Indeed, in some parts of the manuscript the authors use the term dormant NSCs as slowly dividing. It is now considered that dormant NSCs cannot be labeled with BrdU and the term qNSC refers to this type of cells to distinguish them from the activated slowly-dividing aNSCs.

In this revised version, we are only using the term LRC in the entire manuscript. We have also removed the term “dormant NSCs” as suggested by the reviewer.

Reviewer Figure 2: (a-b) Representative confocal images of SVZ immunostained for PCNA and BrdU from the control (a) and demyelinated (b) hemispheres at dorsal, lateral and ventral levels (top, middle and bottom panels respectively) at 7DPI. The absence of PCNA reactivity within the BrdU+ LRC population is indicative of quiescent NSCs (indicated by arrows) and its expression with the BrdU+ population denotes mitotically active cells (indicated by arrowheads). Scale bar represents 25µm.

- In Figure 6, the pictures for the vehicle and Lyso treatments are from different hemispheres and, therefore, it makes sense that the lateral ventricle in the photographs is either to the left

or to the right of the SVZ tissue. But it is very confusing finding the same situation in Figure 5, before the experimental scheme. And, even more, that the lateral ventricles are inverted with respect to the panels in Figure 6.

We thank the reviewer for this observation. To address this issue, we have now included the experimental scheme when we first introduce the lysolecithin injection paradigm (in Figure 4). We have also corrected the images of the SVZ from CTRL (NaCl) and Lysolecithin hemispheres and keep it consistent with the scheme shown in Figures 4 and 6.

Reviewer #3 (Remarks to the Author):

Authors have answered all main points and improved the paper substantially.

We thank the reviewer for their interest in our work.

References

1. Kawaguchi, D., Furutachi, S., Kawai, H., Hozumi, K. & Gotoh, Y. Dll1 maintains quiescence of adult neural stem cells and segregates asymmetrically during mitosis. *Nature communications* **4**, 1880 (2013).
2. Karpowicz, P. *et al.* E-Cadherin regulates neural stem cell self-renewal. *The Journal of neuroscience : the official journal of the Society for Neuroscience* **29**, 3885-3896 (2009).
3. Bicker, F. *et al.* Neurovascular EGFL7 regulates adult neurogenesis in the subventricular zone and thereby affects olfactory perception. *Nature communications* **8**, 15922 (2017).
4. Hu, X.L. *et al.* Persistent Expression of VCAM1 in Radial Glial Cells Is Required for the Embryonic Origin of Postnatal Neural Stem Cells. *Neuron* **95**, 309-325 e306 (2017).
5. Imayoshi, I., Sakamoto, M., Yamaguchi, M., Mori, K. & Kageyama, R. Essential roles of Notch signaling in maintenance of neural stem cells in developing and adult brains. *The Journal of neuroscience : the official journal of the Society for Neuroscience* **30**, 3489-3498 (2010).
6. Furutachi, S. *et al.* Slowly dividing neural progenitors are an embryonic origin of adult neural stem cells. *Nature neuroscience* **18**, 657-665 (2015).
7. Mich, J.K. *et al.* Prospective identification of functionally distinct stem cells and neurosphere-initiating cells in adult mouse forebrain. *Elife* **3**, e02669 (2014).
8. Codega, P. *et al.* Prospective identification and purification of quiescent adult neural stem cells from their in vivo niche. *Neuron* **82**, 545-559 (2014).
9. Reeve, R.L., Yammine, S.Z., Morshead, C.M. & van der Kooy, D. Quiescent Oct4+ Neural Stem Cells (NSCs) Repopulate Ablated Glial Fibrillary Acidic Protein+ NSCs in the Adult Mouse Brain. *Stem cells* **35**, 2071-2082 (2017).
10. Morshead, C.M. & van der Kooy, D. Postmitotic death is the fate of constitutively proliferating cells in the subependymal layer of the adult mouse brain. *The Journal of neuroscience : the official journal of the Society for Neuroscience* **12**, 249-256 (1992).
11. Luskin, M.B. Restricted proliferation and migration of postnatally generated neurons derived from the forebrain subventricular zone. *Neuron* **11**, 173-189 (1993).
12. Morshead, C.M., Craig, C.G. & van der Kooy, D. In vivo clonal analyses reveal the properties of endogenous neural stem cell proliferation in the adult mammalian forebrain. *Development* **125**, 2251-2261 (1998).

13. Lois, C. & Alvarez-Buylla, A. Long-distance neuronal migration in the adult mammalian brain. *Science* **264**, 1145-1148 (1994).
14. Maslov, A.Y., Barone, T.A., Plunkett, R.J. & Pruitt, S.C. Neural stem cell detection, characterization, and age-related changes in the subventricular zone of mice. *The Journal of neuroscience : the official journal of the Society for Neuroscience* **24**, 1726-1733 (2004).
15. Dulken, B.W., Leeman, D.S., Boutet, S.C., Hebestreit, K. & Brunet, A. Single-Cell Transcriptomic Analysis Defines Heterogeneity and Transcriptional Dynamics in the Adult Neural Stem Cell Lineage. *Cell reports* **18**, 777-790 (2017).
16. Klingener, M. *et al.* N-cadherin promotes recruitment and migration of neural progenitor cells from the SVZ neural stem cell niche into demyelinated lesions. *The Journal of neuroscience : the official journal of the Society for Neuroscience* **34**, 9590-9606 (2014).

REVIEWERS' COMMENTS:

Reviewer #1 (Remarks to the Author):

Authors have addressed all my concerns. Even if the flow and clarity of the manuscript could still be improved, I don't see any valid reason for not accepting this nevertheless interesting manuscript for publication in Nature Comm.

I'd like to make just one comment. The miss labelling of the axes in the LRC experiments is very unfortunate. Quantification of LRCs was the main concern of my original review and I am therefore very surprised that such a mistake breaks into the revised version of the manuscript. This lack of rigor is illustrated by several other comments made in the first or second rounds of reviews (e.g. representative pictures for LacZ, missing information in the mat and methods, confusion in the labelling of experimental groups in the figure...). I just want to stress to the authors the importance of performing a careful and rigorous editing of their manuscript before submission, as I believe this not to be part of the reviewer's duties.

Reviewer #2 (Remarks to the Author):

The authors have successfully addressed my concerns.

Point-by-point response to reviewer' comments:

Reviewer #1 (Remarks to the Author)

Authors have addressed all my concerns. Even if the flow and clarity of the manuscript could still be improved, I don't see any valid reason for not accepting this nevertheless interesting manuscript for publication in Nature Comm. I'd like to make just one comment. The miss labelling of the axes in the LRC experiments is very unfortunate. Quantification of LRCs was the main concern of my original review and I am therefore very surprised that such a mistake breaks into the revised version of the manuscript. This lack of rigor is illustrated by several other comments made in the first or second rounds of reviews (e.g. representative pictures for LacZ, missing information in the mat and methods, confusion in the labelling of experimental groups in the figure...). I just want to stress to the authors the importance of performing a careful and rigorous editing of their manuscript before submission, as I believe this not to be part of the reviewer's duties.

We thank the reviewer for their continued interest in our work and apologize for the earlier issues. We have thoroughly checked over the entire manuscript and made sure all the figures and axes are labelled accurately.

Reviewer #2 (Remarks to the Author)

The authors have successfully addressed my concerns.

We thank the reviewer for their interest in our work.